# THE THEORETICAL BENEFITS AND LIMITATIONS OF LATENT CHAIN-OF-THOUGHT REASONING

## ABSTRACT

Recent advances in Latent Chain-of-Thought (Latent CoT) have gained significant attention, yet these models exhibit inconsistent performance across tasks and lack a rigorous theoretical understanding. Our contributions are threefold: (1) We theoretically characterize the fundamental exploration-execution trade-off. We prove that CoT's discrete, symbolic nature forces it into a high-certainty regime, guaranteeing computational fidelity but causing premature commitment that cripples exploration. Conversely, we show that Latent CoT's continuous representation enables robust exploration but is also the direct cause of its failure on computational tasks by amplifying noise. (2) We introduce the Symbolic Index—a measure of a model's decisional certainty—as the core mechanism governing this trade-off. Our unified framework proves that this single, quantifiable metric causally explains the contrasting behaviors of both paradigms, offering a principled way to analyze and design reasoning systems. (3) We prove that curriculum learning is a theoretically grounded and necessary method for training Latent CoT models. We show that without it, training is guaranteed to fail due to a fundamental distributional mismatch, confirming that the staged approach is essential for convergence. This work provides concrete design principles for next-generation reasoning architectures, suggesting a shift from a binary choice between architectures to designing adaptive systems that can dynamically regulate their decisional certainty.

## 1 INTRODUCTION

Large language models (LLMs) (Guo et al. (2025); OpenAI (2025); DeepSeek-AI (2025)) have significantly advanced reasoning tasks (Liu et al. (2025)). A primary method is Chain-of-Thought (CoT) (Wei et al. (2022); Sun et al. (2023)), which verbalizes intermediate steps but often suffers from inefficiency due to excessive length (Hong et al. (2025); Yue et al. (2025)). To overcome this, implicit reasoning has been explored (Ye et al. (2025)), where reasoning occurs internally without generating explicit traces. This is achieved through various mechanisms, including token-level manipulation (Tack et al. (2025); Sun et al. (2025)), trajectory-level optimization (Cheng & Van Durme (2024); Hao et al. (2024a)), latent state refinement (Deng et al. (2023); Kong et al. (2025b)), and signal-guided control (Herel & Mikolov (2024); Goyal et al. (2024); Pfau et al. (2024); Wang et al. (2024)). By operating in the model's continuous representation space, this "silent" reasoning reduces complexity (Chen et al. (2025); Zhu et al. (2025b)), bypasses autoregressive bottlenecks, and enables more diverse and parallelizable reasoning paths (Hao et al. (2024a); Zhang et al. (2025a); Xu et al. (2025a); Gozeten et al. (2025))

However, the two paradigms of explicit reasoning (CoT) and implicit reasoning (Latent CoT) exhibit starkly contrasting and unpredictable performance characteristics across different tasks. As shown in Table 1, explicit CoT excels on tasks requiring precise computation (e.g., GSM8k math problems) with 42.9% accuracy but performs relatively poorly on tasks needing flexible exploration (e.g., ProsQA) at 77.5%. In contrast, implicit reasoning methods based on latent state modeling significantly outperform standard CoT on exploratory reasoning tasks (97.0% vs. 77.5%) but struggle on computational tasks. Even more puzzling, implicit reasoning models show extreme training instability: removing curriculum learning from Coconut leads to catastrophic performance drops (from 99.8% to 52.4%).

Table 1: Performance of discrete (CoT) and latent (Coconut) reasoning models reveals a puzzling trade-off. CoT excels at precise computation (GSM8k) but struggles with flexible reasoning (ProsQA). Conversely, Latent CoT thrives on flexibility but fails at computation and is highly sensitive to its training curriculum. *Results from Hao et al. (2024b).

| Method | GSM8k | | ProntoQA | | ProsQA | |
|---|---|---|---|---|---|---|
| | Acc. (%) | # Tokens | Acc. (%) | # Tokens | Acc. (%) | # Tokens |
| CoT | $42.9 \pm 0.2$ | 25.0 | $98.8 \pm 0.8$ | 92.5 | $77.5 \pm 1.9$ | 49.4 |
| No-CoT | $16.5 \pm 0.5$ | 2.2 | $93.8 \pm 0.7$ | 3.0 | $76.7 \pm 1.0$ | 8.2 |
| COCONUT | $34.1 \pm 1.5$ | 8.2 | $99.8 \pm 0.2$ | 9.0 | $97.0 \pm 0.3$ | 14.2 |
| - w/o curriculum | $14.4 \pm 0.8$ | 8.2 | $52.4 \pm 0.4$ | 9.0 | $76.1 \pm 0.2$ | 14.2 |

This contradictory phenomenon indicates a fundamental architectural trade-off. This paper aims to theoretically explain these performance trade-offs, characterize the essential demands of different task types, and provide principled guidance for designing next-generation reasoning architectures.

To address this challenge, we adopt a systematic analysis based on information theory and decision theory. Our approach is to first deconstruct the performance of each paradigm from first principles, and then build a unified theory from our findings. This allows us to move beyond empirical observations and establish a causal understanding of their behaviors.

Our work makes three contributions:

(1) **We theoretically characterize the fundamental exploration-execution trade-off.** We prove that CoT's high-certainty regime guarantees perfect computational fidelity but leads to vanishing exploration due to early path commitment. In contrast, we show that Latent CoT's low-certainty regime enables robust exploration but is also the *direct cause* of its failure on computational tasks like GSM8k by amplifying noise and preventing the precision required for sequential logic.

(2) **Our unified framework, centered on the Symbolic Index, provides concrete design principles for next-generation reasoning architectures.** It shows how to dynamically regulate decisional certainty based on task requirements, offering a principled approach to building adaptive systems that seamlessly transition between high-fidelity execution and robust exploration.

(3) **We prove that Coconut's curriculum learning is a theoretically grounded and necessary method for training Latent CoT models.** We show that without this curriculum, training is guaranteed to fail due to a fundamental distributional mismatch, while our analysis confirms that the staged approach ensures convergence to an effective reasoning policy.

Our analysis systematically deconstructs this trade-off. We first establish a theoretical foundation by linking Latent CoT to the Conditional Information Bottleneck (Section 4.1). We then prove the superiority of Latent CoT in exploration (Section 4.2) and the fragility of its symbolic computation (Section 4.3). To unify these findings, we introduce the Symbolic Index (Section 4.4) as the core regulatory mechanism. Finally, we prove the theoretical necessity of curriculum learning for stable training (Section 4.5).

## 2 RELATED WORK

### 2.1 EXPLORATORY DIVERSIFICATION AND DECISIONAL CERTAINTY.

Unlike explicit reasoning, which is constrained to a single discrete path, implicit reasoning leverages continuous latent spaces to achieve **exploratory diversification**. Research in this area enables models to explore multiple reasoning trajectories in parallel, enhancing robustness on complex problems. Techniques include sampling latent trajectories (Chen et al. (2024)), using probabilistically weighted concepts (Zhang et al. (2025b)), and injecting continuous tokens to enrich the search space (Xu et al. (2025b;a); Gozeten et al. (2025)). These empirical successes demonstrate the value of exploratory capabilities. Our work provides a theoretical foundation for these findings, proving in Section 4.2 that this capability stems from maintaining a low-certainty state (a low **Symbolic Index**) .

## 2.2 Trajectory-Level Latent Optimization and Progressive Internalization.

Another line of research focuses on **Trajectory-Level Latent Optimization**, compressing entire explicit reasoning chains into continuous representations. Early works aimed to maintain semantic fidelity by anchoring latent states to explicit steps (Cheng & Van Durme (2024); Liu et al. (2024); Shen et al. (2025b)). More recent efforts improve efficiency through dynamic compression or adaptive control (Zhang et al. (2025a); Ma et al. (2025); Tan et al. (2025); Wang et al. (2025a)). A particularly important strategy is progressive internalization, where explicit steps are gradually replaced by latent thoughts via curriculum learning (Deng et al. (2024); Hao et al. (2024a); Shen et al. (2025a)) or internal iterations (Zeng et al. (2025); Ruan et al. (2025)). These methods, especially those using a curriculum, provide the empirical motivation for our analysis. Our work offers a theoretical explanation, proving in Section 4.5 that such progressive training is a necessary mechanism to overcome a fundamental distributional mismatch and ensure convergence.

## 2.3 Signal-Guided Control and Internal State Analysis.

A final stream of research influences the model's internal computation process in a more fine-grained manner. One approach is **Signal-Guided Control**, which inserts specialized, non-text-producing tokens to steer internal reasoning (Herel & Mikolov (2024); Goyal et al. (2024); Zelikman et al. (2024); Pfau et al. (2024); Wang et al. (2024)). Another approach involves **Internal State Analysis**, which uses techniques like probing or distillation to find mechanistic evidence of implicit reasoning, such as discovering encoded reasoning trees in attention patterns (Deng et al. (2023); Hou et al. (2023); Wang et al. (2025b); Yu et al. (2024)). Both approaches highlight the centrality of internal states. Our Symbolic Index framework provides a unified theoretical perspective for these findings, offering a computable metric for these states and proving that variance in their certainty is the root cause of the exploration-execution trade-off (Section 4.4).

## 3 Preliminaries

We consider supervised learning for reasoning tasks, where the input is a random variable $X$ from which instances $x$ are drawn. Each input $x$ and answer $y$ are connected by a chain-of-thought $S = (s_1, \ldots, s_M)$. We compare two paradigms for modeling this process: Chain-of-Thought (CoT) and Latent CoT, the latter trained via the Coconut curriculum.

**Chain-of-Thought (CoT)** CoT models autoregressively generate discrete reasoning steps:

$$p_\theta(S|x) = \prod_{k=1}^{M} p_\theta(s_k|x, S^{(1\ldots k-1)}).$$

At inference, token selection commits to a single path—enabling precise computation but risking early missteps (Mohtashami et al. (2025); Yu et al. (2025); Xu et al. (2024); Emmons et al. (2025)).

**Latent Chain-of-Thought (Latent CoT)** Latent CoT iterates in continuous space, producing latent states $H = (h_1, \ldots, h_M), h_k \in \mathbb{R}^d$:

$$h_k = f_\theta(h_{k-1}).$$

Each $h_k$ encodes multiple potential reasoning paths, supporting exploration but accumulating noise over steps (Kong et al. (2025a); Zhu et al. (2025a); Su et al. (2025); Orlicki (2025)).

**Coconut Training (Hao et al. (2024b))** Since direct supervision on $H$ is infeasible, Coconut uses a curriculum. At stage $k$, it learns to compress the prefix $S^{(1\ldots k)}$ into a latent state $h_k$ to predict the suffix $S^{(k+1\ldots M)}$:

$$\mathcal{L}_{\text{Coconut}}^{(k)}(\theta) = \mathbb{E}_{p(x,S)}[-\log p_\theta(S^{(k+1\ldots M)}|h_k, x)].$$

The curriculum progresses from $k = 0$ (pure CoT) to $k = M$ (full latent reasoning), gradually internalizing symbolic steps which bridges the discrete and continous CoT.

# 4 THEORETICAL ANALYSIS

This section provides a deep theoretical analysis of the fundamental differences and trade-offs between CoT and Latent CoT. We begin by revealing the mathematical underpinnings of the Coconut training objective through the lens of Information Bottleneck theory. Subsequently, we directly analyze model performance along two key dimensions: **planning and exploration capability** versus **computational execution precision**. Finally, we identify **decisional certainty**—quantified by the Symbolic Index—as the core mechanism regulating this trade-off, and theoretically establish the necessity of curriculum learning for effective training of Latent CoT models.

## 4.1 DUALITY WITH THE INFORMATION BOTTLENECK

Directly analyzing the behavior of the Coconut-trained model is challenging due to its staged, implicit compression objective. To overcome this, we show that the Coconut curriculum is *mathematically equivalent* to solving a well-studied information-theoretic objective—the Conditional Information Bottleneck Tishby & Zaslavsky (2015). This equivalence allows us to leverage established tools from information theory to rigorously characterize the model's properties.

This duality can be understood intuitively: the Coconut objective at stage $k$ poses an information compression problem. The model must compress the past chain-of-thought, $S^{(1...k)}$, into a single latent vector, $h_k$, with the sole purpose of maximizing its utility for predicting the future chain, $S^{(k+1...M)}$. This is precisely the problem addressed by the Information Bottleneck (IB) principle: finding a compressed "bottleneck" representation of a source (the past) that maximally preserves information about a target (the future). The Coconut training process thus implicitly learns an optimal information compressor. Theorem 1 formalizes this intuition.

**Theorem 1** (Coconut-CIB Duality). *Under the constraint of any finite model capacity, the optimization objective of the Coconut curriculum (Preliminaries 3) at stage $k$ can be rigorously reformulated as a constrained optimization problem. Its Lagrangian dual is precisely the Conditional Information Bottleneck (CIB) problem. Specifically:*

*(1) **Primal Problem:** The optimization process of Coconut is equivalent to solving:*

$$\min_{p(h_k|X)} H(S^{(k+1...M)} \mid h_k, X) \quad s.t. \quad I(h_k; S^{(1...k)} \mid X) \leq R$$

*where $R$ represents the effective information capacity of the encoder.*

*(2) **Dual Problem:** Its dual problem is to solve the CIB objective:*

$$\min_{p(h_k|X)} \left\{ I(h_k; S^{(1...k)} \mid X) - \beta(k)I(h_k; S^{(k+1...M)} \mid X) \right\}$$

*where the trade-off parameter $\beta(k) > 0$ ideally satisfies $\beta(k) \sim \frac{k}{M-k}$.*

*The main challenge is to connect Coconut's operational training loss to a formal information-theoretic objective. The proof achieves this by framing the optimization as a constrained problem and applying Lagrangian duality, revealing that Coconut implicitly solves the Conditional Information Bottleneck. Proof in Appendix A.2.*

## 4.2 LATENT COT EXCELS IN EXPLORATION

We model reasoning as traversal over a decision DAG $Q := (G, v_{\text{start}}, V_{\text{target}})$. At any node $v$, let $N_{\text{valid}}(v)$ be the set of valid next steps. An ideal explorer uses a uniform prior $q_{\text{PR}}(u \mid v) = \frac{1}{|N_{\text{valid}}(v)|}\mathbb{I}[u \in N_{\text{valid}}(v)]$. We measure deviation from this ideal via $D_{\text{KL}}(q_{\text{PR}}\|p)$, where $p$ is the model's next-step distribution: low KL divergence implies broad exploration; high KL divergence indicates overconfidence and premature commitment.

Our analysis first focuses on the intrinsic properties of a **single, coherent reasoning path**, which constitutes the fundamental building block of the CoT paradigm. Understanding the generative process of one such trajectory is essential for dissecting the core mechanism of execution. To formalize this, we model the high-certainty behavior characteristic of generating a valid computational step.

**Assumption 1** ($\kappa$-Concentrated Distribution for CoT). *At a decision point with $B$ options, we model the generative distribution of a single CoT step, $p_{CoT}$, as being drawn from a Dirichlet prior with a large concentration parameter $\kappa = \sum_i \alpha_i$. A large $\kappa$ yields a sharply peaked distribution, reflecting the high decisional certainty required for deterministic, high-fidelity execution.*

Under this assumption, CoT inevitably collapses its distribution at each step:

**Theorem 2** (Exploration Deficiency of CoT). *As $\kappa \to \infty$, the entropy $H(p_{CoT}) \to 0$, and*

$$D_{KL}(q_{PR}\|p_{CoT}) = \frac{B-1}{B}\log\kappa - \log B - \frac{(B-1)\log c}{B} + \mathcal{O}\left(\frac{1}{\kappa}\right),$$

*for a constant $c > 0$, implying that $D_{KL} \to \infty$ as certainty $\kappa \to \infty$.*

*Proof in Appendix A.3.* **Remark on Sampling-Based Methods.** Our analysis governs the generation of a single reasoning chain. While ensemble methods like Self-Consistency (Wang et al. (2023)) introduce exploration by sampling multiple chains, the computational integrity of each *individual* chain still hinges on the high-certainty, high-$\mathcal{I}_S$ commitments that define CoT's execution-focused nature. Therefore, our core results remain fundamental.

This explains CoT's poor performance on exploratory tasks like ProsQA (77.5% vs. Latent CoT's 97.0% in Table 1): early commitment prevents recovery from missteps.

In contrast, Latent CoT's training objective—dual to the Conditional Information Bottleneck (Theorem 1)—regularizes against overconfidence, ensuring sustained exploration:

**Theorem 3** (Exploration Capability Guarantee of Latent CoT). *There exist constants $\delta \in (0,1)$ and finite $c$ such that*

$$D_{KL}(q_{PR}\|p_{Coconut}) \leq -\frac{1}{2}\log\delta - c.$$

*Proof in Appendix A.4.* This bound guarantees that Latent CoT maintains a non-degenerate distribution over options, enabling robust exploration—complementing CoT's strength in execution, which we analyze next.

### 4.3 LATENT COT'S FRAGILITY IN SYMBOLIC COMPUTATION

While Latent CoT's continuous state space supports robust exploration (§4.2), it is a liability for tasks requiring high-fidelity symbolic reasoning, such as GSM8k. Unlike CoT's discrete symbolic operations, Latent CoT's continuous state representations are inherently vulnerable to noise accumulation, which undermines precise step-by-step computation.

To formalize this, we analyze the effect of small internal errors, termed **sub-decisional perturbations**—noise that corrupts a model's internal state but is insufficient to alter the immediate output.

**Definition 1** (Sub-decisional Perturbation). *Let $l_k = l_k^* + \epsilon_k$ be the logit vector at step $k$. A perturbation $\epsilon_k$ is sub-decisional if*

$$\mathrm{argmax}(l_k^* + \epsilon_k) = \mathrm{argmax}(l_k^*).$$

This concept highlights a key architectural divergence. CoT models are inherently robust to such perturbations due to their discrete generation process.

**Theorem 4** (Symbolic Integrity of CoT). *Let $S^* = (s_1^*, \ldots, s_M^*)$ be the noise-free CoT trajectory and $\hat{S}$ the trajectory under sub-decisional perturbations. Then*

$$\mathbb{P}[\hat{s}_M \neq s_M^*] = 0.$$

Theorem 4 establishes CoT's immunity to this class of noise. The underlying mechanism is a **discretization-reset** process: at each step, the model projects its continuous hidden state to a discrete token via an "argmax" operation. This act of discretization discards any sub-decisional noise present in the hidden state. The subsequent reasoning step thus begins from a clean, noise-free symbolic representation, preventing error propagation.

In stark contrast, Latent CoT lacks this reset mechanism. Its continuous state $h_k$ is passed directly to the next step, carrying forward any perturbation. For a model with transition function $f_\theta$ (Lipschitz constant $L_F$) and i.i.d. noise $\epsilon_h^{(k)} \sim \mathcal{N}(0, \sigma_h^2 I_d)$, this error propagation is quantifiable:

**Theorem 5** (Compounding Error in Latent Computation). *Let $E_M = h_M - h_M^*$. Then for $M \geq 1$,*

$$\mathbb{E}[\|E_M\|^2] = \frac{1 - L_F^{2M}}{1 - L_F^2} \cdot d\sigma_h^2 > 0.$$

*Proof in Appendix A.5.* The term $\frac{1 - L_F^{2M}}{1 - L_F^2}$ demonstrates that the expected squared error grows with the number of reasoning steps $M$. If the model's transition function is expansive ($L_F > 1$), the error compounds exponentially; even for stable functions ($L_F \leq 1$), error still accumulates. This mathematical result directly explains Latent CoT's performance degradation on precision-sensitive tasks like GSM8k. The accumulation of even small perturbations over a long reasoning chain corrupts the final state, undermining the integrity required for symbolic computation and complementing the dichotomy analyzed in §4.2.

### 4.4 UNIFYING THE TRADE-OFF VIA THE SYMBOLIC INDEX

The analyses in §4.2 and §4.3 reveal a dichotomy: CoT excels at execution but lacks exploration, while Latent CoT does the opposite. Here, we unify these behaviors through a single, quantifiable principle: the degree of decisional certainty at each reasoning step. To formalize this, we introduce the **Symbolic Index** ($\mathcal{I}_S$), a measure of how committed the model is to its top choice.

**Definition 2** (Symbolic Index $\mathcal{I}_S$). *At a decision point with a vocabulary of valid tokens $\mathcal{V}$, let $p(u \mid h, x)$ be the model's output distribution. The Symbolic Index is the probability of the most likely token:*

$$\mathcal{I}_S = \max_{u \in \mathcal{V}} p(u \mid h, x) \in [1/|\mathcal{V}|, 1].$$

A high $\mathcal{I}_S$ ($\approx 1$) signifies high certainty, characteristic of CoT's discrete commitments. A low $\mathcal{I}_S$ signifies distributed consideration over multiple options, characteristic of Latent CoT.

First, we establish the direct link between this certainty and robustness against computational noise. The mechanism for this robustness is the separation between the top two choices, which we term the Logit Decision Margin.

**Definition 3** (Logit Decision Margin $\Delta_l$). *Let $l_{i^*}$ and $l_{j^*}$ be the logits of the most likely and second-most likely tokens, respectively. The margin is their difference: $\Delta_l = l_{i^*} - l_{j^*}$.*

The following theorem proves that high certainty guarantees a large decision margin, making the model resilient to perturbations.

**Theorem 6** (Symbolic Stability Theorem). *The Logit Decision Margin $\Delta_l$ is lower-bounded by the Symbolic Index $\mathcal{I}_S$ as follows:*

$$\Delta_l \geq \log \left( \frac{\mathcal{I}_S}{1 - \mathcal{I}_S} \right).$$

*Proof in Appendix A.8.* This theorem establishes a direct relationship: higher decisional certainty guarantees a larger protective margin. For instance, a high-certainty CoT with $\mathcal{I}_S = 0.99$ has a margin $\Delta_l \geq \log(99) \approx 4.6$, making its decision robust to significant noise. In contrast, a low-certainty Latent CoT with $\mathcal{I}_S = 0.6$ has a margin of only $\Delta_l \geq \log(1.5) \approx 0.4$, rendering it vulnerable to small perturbations. This explains CoT's noise immunity and symbolic integrity (Theorem 4).

However, this stability comes at a direct and quantifiable cost to exploration. High certainty forces the model's distribution away from the uniform ideal required for unbiased exploration. The next theorem formalizes this trade-off.

**Theorem 7** (Exploration-Execution Trade-off Theorem). *For a decision with $B$ valid options, the KL divergence from the ideal uniform exploration prior $q_{PR}$ is lower-bounded by:*

$$D_{KL}(q_{PR}\|p) \geq \log B + \mathcal{I}_S \log(\mathcal{I}_S) + (1 - \mathcal{I}_S) \log \left( \frac{1 - \mathcal{I}_S}{B - 1} \right).$$

*This bound is minimized when $\mathcal{I}_S = 1/B$ and grows as $\mathcal{I}_S \to 1$.*

*Proof in Appendix A.9.* This inequality proves that gaining execution stability by increasing $\mathcal{I}_S$ *necessarily* degrades exploration capability by increasing the divergence from a uniform policy. It quantifies the inherent tension between committing to a single path and keeping options open, explaining CoT's exploration deficiency (Theorem 2).

Together, these results establish $\mathcal{I}_S$ as the core regulator of the trade-off, unifying the behaviors of both paradigms:

(1) **CoT** operates in a high-$\mathcal{I}_S$ regime. This ensures a large decision margin for robust execution (Theorems 6, 4) but at the cost of poor exploration (Theorem 7).

(2) **Latent CoT** is trained into a low-$\mathcal{I}_S$ regime, enabling broad exploration (Theorem 3). However, this low certainty implies a small decision margin, making it vulnerable to sub-decisional noise. This vulnerability, combined with compounding error (Theorem 5), leads to failure on precision-sensitive tasks.

This framework shifts the design focus from a binary choice of architecture to the problem of **managing decisional certainty**. The path forward may lie in adaptive systems that can dynamically regulate their $\mathcal{I}_S$ based on task demands—exploring broadly when needed and committing precisely during execution.

### 4.5 WHY CURRICULUM LEARNING WORKS

Our analysis explains a trained Latent CoT model's behavior but not its training instability without a curriculum (Table 1). Here, we prove that curriculum learning is theoretically necessary to overcome a fundamental challenge: the absence of ground-truth latent thoughts. We frame this as an **Imitation Learning (IL)** problem, where a student policy $P_{\hat{\theta}}(h|x)$ (the Latent CoT model) must learn an unobserved expert policy $P_{\theta^*}(h|x)$ that generates optimal, reasoning-encoded latent states. The goal is to learn $\hat{\theta}$ that maximizes the success rate $\mathbb{E}_{x,h \sim P_{\hat{\theta}}}[V(h)]$, where $V(h) = 1$ indicates a correct final answer.

**The Failure of Direct Training: A Distributional Mismatch.** Without a curriculum, the model must learn from its own generated latent states. An untrained model, however, has no incentive to perform multi-step reasoning. Instead, it learns to generate "shortcut" latent states that merely capture superficial input-output correlations (e.g., mapping keywords in the question directly to an answer token). The model is therefore trained on samples from a biased distribution $P_{\text{biased}}(h|x)$ of these non-reasoning states, which is fundamentally different from the true expert distribution $P_{\theta^*}(h|x)$. The following theorem proves that learning from this mismatched distribution guarantees failure.

**Theorem 8** (Provable Failure of Training without Curriculum). *Let $D_{nc}$ be a dataset of latent states drawn from a biased, non-reasoning distribution $P_{biased} \neq P_{\theta^*}$. The model $P_{\hat{\theta}_{MLE}}$ trained on this data via Maximum Likelihood Estimation is bounded away from the expert policy, such that:*

$$\mathbb{E}_{D_{nc}}[D_{KL}(P_{\theta^*} \| P_{\hat{\theta}_{MLE}})] \geq C > 0,$$

*for some constant $C$, irrespective of dataset size. Consequently, the model's success rate remains strictly suboptimal.*

*Proof in Appendix A.6.* This theorem shows that training on "shortcut" latent states creates a permanent gap between the learned model and the ideal reasoning model. No amount of data can fix this fundamental distributional mismatch, explaining the performance collapse in Table 1.

**The Success of Curriculum Learning: Bridging the Distributional Gap.** Coconut's curriculum systematically resolves this issue. At each stage $k$, the model is forced to generate a latent state $h_k$ that compresses the *expert* reasoning prefix $S^{(1...k)}$ to predict the *expert* suffix $S^{(k+1...M)}$. This procedure effectively provides supervised samples of latent thoughts that are grounded in correct reasoning steps. In the IL framework, this is equivalent to drawing samples directly from the true expert distribution $P_{\theta^*}(h|x)$. By transforming the problem into a well-posed supervised learning task, convergence is assured.

**Theorem 9** (Provable Success of Training with Curriculum). *Under standard statistical learning conditions, MLE on a dataset $D_c$ of size $n$ generated via the curriculum yields a model $\hat{\theta}$ whose success rate approaches that of the expert:*

$$SuccessRate(\hat{\theta}) \geq SuccessRate(\theta^*) - \mathcal{O}\left(\sqrt{\frac{d \log n + \log(1/\delta)}{n}}\right),$$

*with high probability $1 - \delta$. The performance gap vanishes as the dataset size $n \to \infty$.*

*Proof in Appendix A.7.* This theorem confirms that by using the explicit CoT as a scaffold to generate valid training data, the curriculum ensures that the student model can provably converge to the optimal expert policy.

In conclusion, these theorems establish that curriculum learning is not merely a helpful heuristic but a **theoretically necessary mechanism**. It resolves the fatal distributional mismatch that dooms direct training, making it an essential component for building effective Latent CoT models.

## 5 EXPERIMENTS

We conduct two sets of experiments to empirically validate the theoretical framework presented in Section 4.4: (1) visualizing the Symbolic Index ($\mathcal{I}_S$) to observe internal decisional certainty, and (2) evaluating model robustness under computational noise to confirm error propagation and stability predictions.

**Experimental Setup** We compare a standard CoT model with a Latent CoT model (Coconut), both based on GPT-2 (124M). We use two benchmarks: **GSM8k** Cobbe et al. (2021) for precise computation, and **ProsQA** Hao et al. (2024b) for exploration. Full implementation and training details are in Appendix A.1.

### 5.1 VISUALIZING THE SYMBOLIC INDEX ($\mathcal{I}_S$)

To empirically estimate the model's internal decisional certainty, we plot the maximum token probability (i.e., $\mathcal{I}_S$) from the renormalized nucleus-sampling distribution ($p = 0.95$) at each reasoning step.

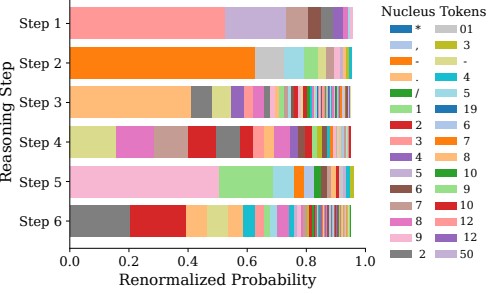 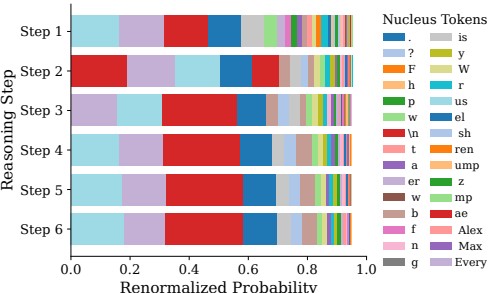

Figure 1: Symbolic Index ($\mathcal{I}_S$) over reasoning steps on GSM8k for Latent CoT (Coconut). The model consistently operates in a low-$\mathcal{I}_S$ regime.

Figure 2: Symbolic Index ($\mathcal{I}_S$) over reasoning steps on ProsQA for Latent CoT (Coconut). $\mathcal{I}_S$ remains low, enabling exploration.

As depicted in Figures 1 and 2, the Coconut model consistently maintains a low $\mathcal{I}_S$ regime across both GSM8k and ProsQA.

- On **ProsQA** (Figure 2), this low $\mathcal{I}_S$ is beneficial, confirming its robust exploration capabilities as predicted by Theorem 3 and the Exploration-Execution Trade-off (Theorem 7).
- On **GSM8k** (Figure 1), the model's inability to increase its $\mathcal{I}_S$ implies a small decision margin, leaving it vulnerable to internal noise. This aligns with the Symbolic Stability Theorem (Theorem 6) and the Compounding Error Theorem (Theorem 5), predicting its fragility in precision-sensitive tasks.

## 5.2 ROBUSTNESS TO COMPUTATIONAL NOISE

We evaluate the models' resilience to internal computational noise by injecting Gaussian noise $\mathcal{N}(0, \sigma^2 I_d)$ into their hidden states during the reasoning process. For Latent CoT, noise is added to the latent thought vector $h_k$ at each step. For standard CoT, noise is applied to the pre-output hidden state before token generation.

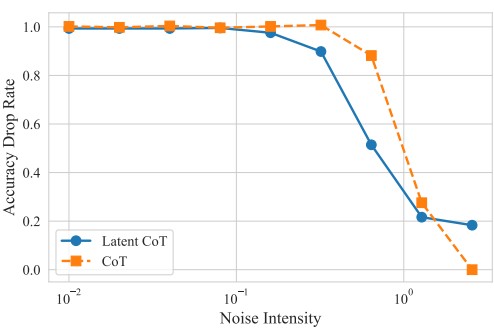
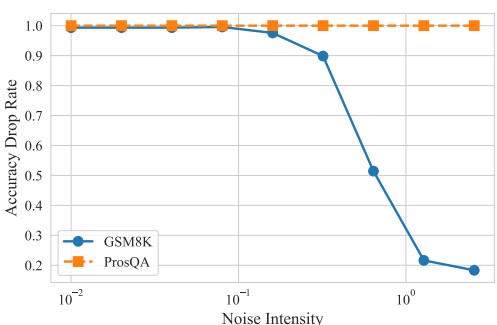

Figure 3: Performance on GSM8k under increasing noise levels, comparing CoT and Latent CoT. Latent CoT degrades progressively, while CoT exhibits an abrupt failure.

Figure 4: Latent CoT's performance on GSM8k vs. ProsQA under noise. It is significantly more fragile on GSM8k.

Figure 3 illustrates the distinct noise robustness profiles on GSM8k:

- **Latent CoT (Coconut)**: Its performance degrades steadily even under low noise levels. This direct empirical observation confirms the theoretical prediction of compounding error in continuous representations (Theorem 5), where internal perturbations accumulate over reasoning steps.
- **Standard CoT**: It maintains high accuracy for a range of noise levels, then experiences an abrupt, catastrophic failure. This "all-or-nothing" behavior is consistent with CoT's high-$\mathcal{I}_S$ regime, where it remains robust until noise exceeds its large Logit Decision Margin (Theorem 6), after which symbolic integrity is lost (Theorem 4).

Figure 4 further highlights Latent CoT's task-dependent fragility. While Latent CoT on GSM8k shows significant degradation with noise, its performance on ProsQA is more resilient. This validates our theoretical distinction: symbolic precision tasks (GSM8k) are highly sensitive to continuous state perturbations due to exact state preservation requirements, whereas exploratory tasks (ProsQA) can tolerate more noise because small internal deviations may still lead to valid (and possibly beneficial) alternative paths, consistent with its inherent low-$\mathcal{I}_S$ exploratory nature.

## 6 CONCLUSION

We introduce the Symbolic Index—a measure of decisional certainty—as a unifying framework that explains the trade-off between exploration and execution in reasoning systems. We prove that CoT's high certainty ensures computational precision but limits exploration, while Latent CoT's low certainty enables robust exploration at the cost of symbolic integrity. We also establish that curriculum learning is theoretically necessary for training Latent CoT, as its absence causes a distributional mismatch that guarantees failure. Our experiments validate these predictions by visualizing internal certainty and measuring noise robustness.

These results suggest a shift from choosing between fixed architectures toward designing adaptive systems that dynamically regulate their Symbolic Index—exploring broadly when decomposing problems and committing precisely during execution. By making decisional certainty a first-class design principle, our work provides a foundation for more flexible and robust reasoning architectures.

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

# A APPENDIX

## A.1 EXPERIMENTAL DETAILS

**Models.** Both CoT and Latent CoT (Coconut) models are based on the smallest GPT-2 variant with 124M parameters. For Latent CoT, we use $M = 6$ latent reasoning steps across all experiments. The latent thought vector $h_k$ has the same dimension as GPT-2's hidden size ($d = 768$).

**Training.** We use the Adam optimizer with learning rate $1e-4$ and weight decay $0.01$. Training is performed on 4 GPUs with mixed-precision (bfloat16 enabled for ProsQA, disabled for GSM8k due to stability). For GSM8k, we train for 25 epochs with batch size 32 and gradient accumulation steps 1; for ProsQA, we train for 30 epochs with batch size 16 and gradient accumulation steps 2. The Coconut curriculum progresses stage-by-stage: for GSM8k, we use 3 epochs per stage up to latent stage 3 (i.e., compressing the first 3 CoT steps into latent thoughts); for ProsQA, we use 4 epochs per stage up to latent stage 6 (full internalization). The base model checkpoint is initialized from standard GPT-2.

**Datasets.** We evaluate on two reasoning benchmarks:

1. **GSM8k**: We use the standard split with a held-out test set of 330 examples. Training and validation use the official training set (7,473 examples) and a 10% validation subset, respectively.
2. **ProsQA**: Following Hao et al. (2024b), we use a test set of 300 examples. The training and validation sets contain 2,000 and 200 examples, respectively.

All reported accuracies are computed on these test sets.

**Symbolic Index Visualization Details.** The visualizations in Figures 1 and 2 are based on representative examples from the respective test sets.

- **Figure 1 (GSM8k):** The plot was generated from the following math reasoning problem: "Janet's ducks lay 16 eggs per day. She eats three for breakfast every morning and bakes muffins for her friends every day with four. She sells the remainder at the farmers' market daily for $2 per fresh duck egg. How much in dollars does she make every day at the farmers' market?"
- **Figure 2 (ProsQA):** The plot was generated from the following logical reasoning problem: "Every shumpus is a yumpus. Every worpus is a yimpus. Every shumpus is a gwompus. Every tumpus is a boompus. Every worpus is a shumpus. Every storpus is a terpus. Max is a yimpus. Every shumpus is a rompus. Every wumpus is a jelpus. Every boompus is a terpus. Fae is a tumpus. Every tumpus is a worpus. Every rompus is a gorpus. Every timpus is a impus. Every jompus is a gerpus. Every boompus is a rompus. Fae is a boompus. Every boompus is a kerpus. Every zumpus is a bompus. Max is a rempus. Every rompus is a kerpus. Max is a impus. Every rempus is a impus. Every wumpus is a yumpus. Every grimpus is a terpus. Every tumpus is a jompus. Every yumpus is a felpus. Every jelpus is a felpus. Every shumpus is a felpus. Every rempus is a timpus. Every storpus is a jompus. Every rompus is a storpus. Every tumpus is a wumpus. Every wumpus is a jompus. Every boompus is a worpus. Fae is a storpus. Every worpus is a jelpus. Every grimpus is a felpus. Every worpus is a yumpus. Every rempus is a zumpus. Every kerpus is a grimpus. Is Fae a gwompus or bompus?"

**Noise Injection Protocol.** During inference, we inject Gaussian noise $\mathcal{N}(0, \sigma^2 I_d)$ as follows:

1. For **CoT**: noise is added to the last hidden state before the LM head, i.e., $h \leftarrow h + \sigma \cdot \epsilon$, where $\epsilon \sim \mathcal{N}(0, I_d)$.
2. For **Latent CoT**: noise is added at every reasoning step, i.e., $h_k \leftarrow f_\theta(h_{k-1}) + \sigma \cdot \epsilon_k$, with independent $\epsilon_k \sim \mathcal{N}(0, I_d)$.

We report accuracy as the exact-match rate on the test set. The "accuracy drop ratio" in Figure 3 is defined as $\text{Acc}(\sigma)/\text{Acc}(0)$.

**Reproducibility.** All experiments use seed 0. Due to the deterministic nature of our evaluation protocol and the use of nucleus sampling with fixed $p = 0.95$ for Symbolic Index estimation, results are fully reproducible given the same model checkpoint.

## A.2 PROOF OF THEOREM 1

**Lemma 1** (Loss and Mutual Information). *Assume the decoder model family $\{p_\theta(S^{(k+1...M)} \mid h_k, X)\}$ is well-specified, i.e., there exists a parameter $\theta^*$ such that $p_{\theta^*}(S^{(k+1...M)} \mid h_k, X) = p(S^{(k+1...M)} \mid h_k, X)$ holds almost everywhere. Further assume $H(S^{(k+1...M)} \mid X) < \infty$. Then,*

$$\min_\theta \mathcal{L}_{Coconut}^{(k)}(\theta) \iff \max_{p(h_k|X)} I(h_k; S^{(k+1...M)} \mid X).$$

*Proof.* Expand the loss function:

$$\mathcal{L}_{\text{Coconut}}^{(k)}(\theta) = \mathbb{E}_{p(X,S,h_k)}[-\log p_\theta(S^{(k+1...M)} \mid h_k, X)]$$

$$= \mathbb{E}_{p(X,h_k)}\left[\mathbb{E}_{p(S^{(k+1...M)}|X,h_k)}[-\log p_\theta(S^{(k+1...M)} \mid h_k, X)]\right]$$

$$= \mathbb{E}_{p(X,h_k)}\left[\mathbb{E}_{p(S^{(k+1...M)}|X,h_k)}\left[-\log p(S^{(k+1...M)} \mid h_k, X) + \log \frac{p(S^{(k+1...M)} \mid h_k, X)}{p_\theta(S^{(k+1...M)} \mid h_k, X)}\right]\right]$$

$$= \mathbb{E}_{p(X,h_k)}\left[H(S^{(k+1...M)} \mid h_k, X) + D_{\text{KL}}(p(\cdot \mid h_k, X)\|p_\theta(\cdot \mid h_k, X))\right].$$

By the well-specified assumption, there exists $w^*$ such that $D_{\text{KL}}(p(\cdot \mid h_k, X)\|p_\theta(\cdot \mid h_k, X)) \to 0$. Therefore,

$$\min_\theta \mathcal{L}_{\text{Coconut}}^{(k)}(\theta) = \min_{p(h_k|X)} H(S^{(k+1...M)} \mid h_k, X).$$

By the definition of conditional mutual information, $I(A; B \mid C) = H(A \mid C) - H(A \mid B, C)$, we have

$$H(S^{(k+1...M)} \mid h_k, X) = H(S^{(k+1...M)} \mid X) - I(h_k; S^{(k+1...M)} \mid X).$$

Since $H(S^{(k+1...M)} \mid X)$ is constant, it follows that

$$\min_{p(h_k|X)} H(S^{(k+1...M)} \mid h_k, X) \iff \max_{p(h_k|X)} I(h_k; S^{(k+1...M)} \mid X).$$

$\square$

We now begin the proof of Theorem 1.

*Proof.* By Lemma 1, minimizing the Coconut loss is equivalent to $\min_{p(h_k|X)} H(S^{(k+1...M)} \mid h_k, X)$. Any physically realizable encoder is constrained by finite model capacity, formally expressed as $I(h_k; S^{(1...k)} \mid X) \leq R$. We construct the Lagrangian:

$$\mathcal{J}(p, \lambda) = H(S^{(k+1...M)} \mid h_k, X) + \lambda \left(I(h_k; S^{(1...k)} \mid X) - R\right), \quad \lambda \geq 0.$$

Using the identity $H(A \mid B, C) = H(A \mid C) - I(B; A \mid C)$, we rewrite:

$$\mathcal{J} = \left[H(S^{(k+1...M)} \mid X) - I(h_k; S^{(k+1...M)} \mid X)\right] + \lambda I(h_k; S^{(1...k)} \mid X) - \lambda R.$$

Ignoring constants $H(S^{(k+1...M)} \mid X)$ and $\lambda R$ during minimization over $p(h_k \mid X)$, we obtain:

$$\min_{p(h_k|X)} \left\{\lambda I(h_k; S^{(1...k)} \mid X) - I(h_k; S^{(k+1...M)} \mid X)\right\}.$$

Defining $\beta = 1/\lambda > 0$ and dividing through by $\lambda$, this becomes the standard CIB objective:

$$\min_{p(h_k|X)} \left\{I(h_k; S^{(1...k)} \mid X) - \beta I(h_k; S^{(k+1...M)} \mid X)\right\}.$$

Denote $I_{\text{past}} = I(h_k; S^{(1...k)} \mid X)$ and $I_{\text{future}} = I(h_k; S^{(k+1...M)} \mid X)$. Any Pareto-optimal encoder lies on the efficiency frontier of the information plane, where the trade-off between retained and predicted information satisfies the first-order condition:

$$\left.\frac{dI_{\text{future}}}{dI_{\text{past}}}\right|_{h_k^*} = \frac{1}{\beta(k)}.$$

We model the frontier as:

$$I_{\text{future}}(I_{\text{past}}; k) = I_{\max}(k)\left(1 - e^{-\alpha(k)\cdot I_{\text{past}}}\right),$$

with parameters motivated by:

$I_{\max}(k) \approx C_2(M - k)$, proportional to the length of the remaining sequence.

$\alpha(k) \approx 1/H(S^{(1...k)} \mid X) \approx 1/(C_1 k + C_0)$, inversely proportional to the entropy of past tokens. Differentiating gives:

$$\frac{dI_{\text{future}}}{dI_{\text{past}}} = I_{\max}(k) \cdot \alpha(k) \cdot e^{-\alpha(k)\cdot I_{\text{past}}}.$$

At optimality, the encoder saturates its bottleneck: $I_{\text{past}} \approx H(S^{(1...k)} \mid X) \approx 1/\alpha(k)$. Substituting:

$$\left.\frac{dI_{\text{future}}}{dI_{\text{past}}}\right|_{h_k^*} \approx I_{\max}(k) \cdot \alpha(k) \cdot e^{-1} \approx \frac{C_2(M - k)}{e(C_1 k + C_0)}.$$

Applying $\beta(k) = (dI_{\text{future}}/dI_{\text{past}})^{-1}$:

$$\beta(k) = e \cdot \frac{C_1 k + C_0}{C_2(M - k)} = e\frac{C_1}{C_2} \cdot \frac{k}{M - k} + O\left(\frac{1}{M - k}\right).$$

Thus, as $k \to M$, the dominant asymptotic behavior is $\beta(k) \sim \frac{k}{M-k}$, revealing that the optimal trade-off scales with the ratio of processed to remaining sequence length. $\square$

### A.3 PROOF OF THEOREM 2

**Lemma 2** (Upper Bound on the Entropy $H(p_{\text{CoT}})$ of the CoT Output Distribution). *Let $\bar{p} = \mathbb{E}[p_{CoT}]$ be the expected next-step distribution over $B$ options, drawn from a Dirichlet prior with parameters $\boldsymbol{\alpha}$ and concentration $\kappa = \sum_{i=1}^{B} \alpha_i$. Under high concentration — i.e., when $\alpha_j \gg \alpha_{i\neq j}$ for some $j$ — the entropy $H(\bar{p})$ vanishes as $\kappa$ grows. Specifically, if $\alpha_j = \kappa - (B-1)c$ and $\alpha_{i\neq j} = c$ for small $c > 0$, then*

$$H(\bar{p}) = O\left(\frac{\log \kappa}{\kappa}\right),$$

*Proof.* By properties of the Dirichlet distribution, the expected probabilities are given by $\bar{p}_i = \alpha_i/\kappa$. We consider a typical unimodal scenario in which the vast majority of the weight concentrates on a single option $j$. We set $\alpha_j = \kappa - (B-1)c$ and $\alpha_{i\neq j} = c$ for all $i \neq j$, where $c$ is a small positive constant.

Under this setting, the expected probabilities are:

$$\bar{p}_j = \frac{\kappa - (B-1)c}{\kappa} = 1 - \frac{(B-1)c}{\kappa}$$

$$\bar{p}_{i\neq j} = \frac{c}{\kappa}$$

We compute the entropy of this distribution:

$$H(\bar{p}) = -\bar{p}_j \log \bar{p}_j - \sum_{i\neq j} \bar{p}_i \log \bar{p}_i$$

$$= -\left(1 - \frac{(B-1)c}{\kappa}\right)\log\left(1 - \frac{(B-1)c}{\kappa}\right) - (B-1)\frac{c}{\kappa}\log\left(\frac{c}{\kappa}\right)$$

We analyze the first term using the Taylor expansion $\log(1 - x) \approx -x$. As $\kappa \to \infty$, we have $\frac{(B-1)c}{\kappa} \to 0$, and thus:

$$-\left(1 - \frac{(B-1)c}{\kappa}\right) \log\left(1 - \frac{(B-1)c}{\kappa}\right) \approx -\left(1 - \frac{(B-1)c}{\kappa}\right)\left(-\frac{(B-1)c}{\kappa}\right) = O\left(\frac{1}{\kappa}\right).$$

Next, we analyze the second term:

$$-(B-1)\frac{c}{\kappa} \log\left(\frac{c}{\kappa}\right) = -(B-1)\frac{c}{\kappa}(\log c - \log \kappa)$$

$$= (B-1)c\frac{\log \kappa}{\kappa} - (B-1)c\frac{\log c}{\kappa} = O\left(\frac{\log \kappa}{\kappa}\right).$$

Combining both terms, as $\kappa$ becomes large, the entropy is dominated by the $O\left(\frac{\log \kappa}{\kappa}\right)$ term. And we have $\lim_{\kappa \to \infty} \frac{\log \kappa}{\kappa} = 0$. $\qquad \square$

We now begin the proof of Theorem 2.

*Proof.* We analyze the almost sure asymptotic behavior of $D_{\mathrm{KL}}(q_{\mathrm{PR}} \| p_{\mathrm{CoT}})$ under the concentrated Dirichlet prior where $\alpha_j = \kappa - (B-1)c$ and $\alpha_{i \neq j} = c$.

By the law of large numbers for Dirichlet distributions, as $\kappa \to \infty$, the random vector $p_{\mathrm{CoT}}$ converges almost surely to its mean:

$$p_{\mathrm{CoT}} \xrightarrow{a.s.} \bar{p} = \left(1 - \frac{(B-1)c}{\kappa}, \frac{c}{\kappa}, \dots, \frac{c}{\kappa}\right).$$

Since the KL divergence is a continuous function of the probability vector (away from the boundary), we have:

$$D_{\mathrm{KL}}(q_{\mathrm{PR}} \| p_{\mathrm{CoT}}) \xrightarrow{a.s.} D_{\mathrm{KL}}(q_{\mathrm{PR}} \| \bar{p}) \quad \text{as } \kappa \to \infty.$$

Now compute $D_{\mathrm{KL}}(q_{\mathrm{PR}} \| \bar{p})$:

$$D_{\mathrm{KL}}(q_{\mathrm{PR}} \| \bar{p}) = \sum_{i=1}^{B} \frac{1}{B} \log \frac{1/B}{\bar{p}_i}$$

$$= -\log B - \frac{1}{B} \sum_{i=1}^{B} \log \bar{p}_i$$

$$= -\log B - \frac{1}{B}\left[\log\left(1 - \frac{(B-1)c}{\kappa}\right) + (B-1)\log\left(\frac{c}{\kappa}\right)\right].$$

Using Taylor expansion $\log(1 - x) = -x + O(x^2)$:

$$D_{\mathrm{KL}}(q_{\mathrm{PR}} \| \bar{p}) = -\log B - \frac{1}{B}\left[-\frac{(B-1)c}{\kappa} + O\left(\frac{1}{\kappa^2}\right) + (B-1)\log c - (B-1)\log \kappa\right]$$

$$= \frac{B-1}{B}\log \kappa - \log B - \frac{(B-1)\log c}{B} + O\left(\frac{1}{\kappa}\right).$$

Therefore, almost surely as $\kappa \to \infty$:

$$D_{\mathrm{KL}}(q_{\mathrm{PR}} \| p_{\mathrm{CoT}}) = \frac{B-1}{B}\log \kappa - \log B - \frac{(B-1)\log c}{B} + \mathcal{O}\left(\frac{1}{\kappa}\right).$$

Since $H(p_{\mathrm{CoT}}) \to 0$ as established in Lemma 2, and $D_{\mathrm{KL}}(q_{\mathrm{PR}} \| p_{\mathrm{CoT}})$ grows logarithmically in $\kappa$, this proves the exploration deficiency of CoT. $\qquad \square$

## A.4 Proof of Theorem 3

**Assumption 2** (Convergence and Compactness of Latent States). *We assume that during inference, the sequence of latent states $\{h_k\}$ converges to a fixed point or enters a compact attracting set as the reasoning step $k$ increases. This is not just a theoretical convenience; it is a behavior we consistently observe empirically. As shown in Figure 5, a PCA visualization of the latent thought embeddings reveals a clear convergent trajectory, where successive states $\{L1, \ldots, L6\}$ move progressively closer within a bounded region. The high explained variance (0.99) of the first two principal components confirms that this 2D projection faithfully represents the dynamics in the original high-dimensional space.*

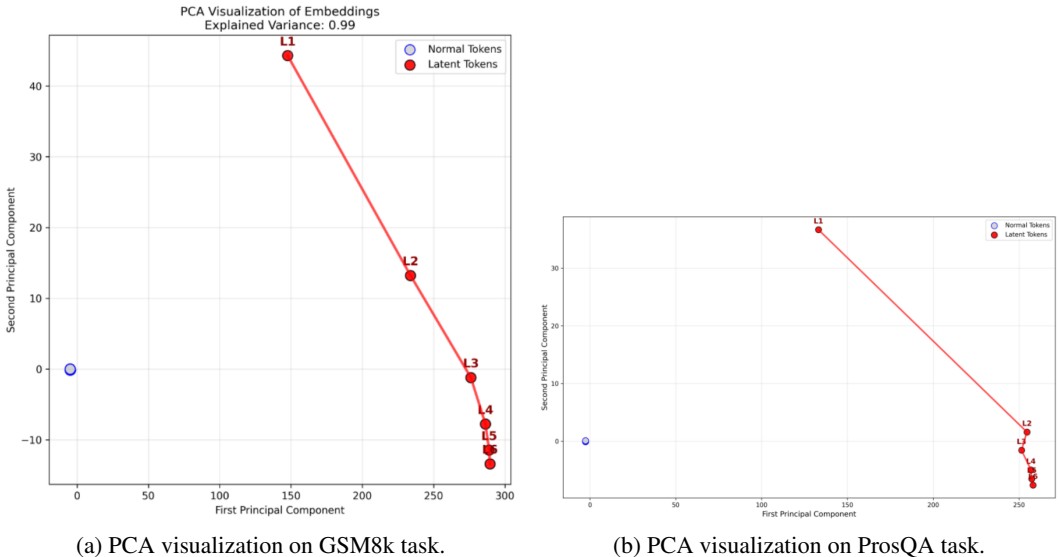

    (a) PCA visualization on GSM8k task.         (b) PCA visualization on ProsQA task.

Figure 5: PCA visualization of latent state embeddings for reasoning trajectories. In both (a) and (b), the latent thoughts (L1-L6) demonstrate a clear convergence, supporting our assumption of a compact, attracting set for latent states. This validates that the convergence dynamic is a general property of the latent reasoning process, not specific to a single task.

**Lemma 3** (Non-Degenerate Output Distribution of Coconut). *For a model trained via the Coconut objective, its optimization can be modeled as solving a Conditional Information Bottleneck (CIB) problem with a finite trade-off parameter $\beta(k) > 0$ (Theorem 1). Under this framework, the model's output probability distribution is non-degenerate: there exists a constant $\delta > 0$ such that for any valid next token $u$ and any reachable latent state $h$, the probability is strictly bounded away from 1:*

$$\sup_{h \in \mathcal{H}} \left( \max_{u \in N_{valid}(v)} p(u \mid h) \right) \leq 1 - \delta,$$

*where $\mathcal{H}$ is the compact latent state space guaranteed by Assumption 2.*

*Proof.* Assume, for contradiction, that $\sup_{h \in \mathcal{H}} \max_u p(u \mid h) = 1$. Since $\mathcal{H}$ is compact and the decoder $p(u \mid h)$ is continuous, there must exist a latent state $h^* \in \mathcal{H}$ and a token $u^*$ such that $p(u^* \mid h^*) = 1$.

At $h^*$, future uncertainty is fully eliminated: $H(S^{(k+1\ldots M)} \mid h^*, X) = 0$, implying that the predictive information $I(h; S^{(k+1\ldots M)} \mid X)$ achieves its theoretical maximum.

By Theorem 1, the Coconut solution corresponds to an optimal solution of a CIB problem. On the information plane $(I_{\text{past}}, I_{\text{future}})$, all optimal solutions must lie on the efficiency frontier, and any optimal point $h_{\text{opt}}$ must satisfy:

$$\left. \frac{dI_{\text{future}}}{dI_{\text{past}}} \right|_{h_{\text{opt}}} = \frac{1}{\beta(k)}.$$

At our hypothetical point $h^*$, $I_{\text{future}}$ reaches its maximum value $I_{\max}(k)$. Under the efficiency frontier model adopted in Theorem 1:

$$I_{\text{future}}(I_{\text{past}}; k) = I_{\max}(k)\left(1 - e^{-\alpha(k)\cdot I_{\text{past}}}\right),$$

which is strictly increasing and strictly concave in $I_{\text{past}}$, with derivative:

$$\frac{dI_{\text{future}}}{dI_{\text{past}}} = I_{\max}(k) \cdot \alpha(k) \cdot e^{-\alpha(k)\cdot I_{\text{past}}}.$$

As $I_{\text{future}} \to I_{\max}(k)$, we must have $I_{\text{past}} \to \infty$, and the derivative $\to 0^+$. If $I_{\text{future}}$ were to reach $I_{\max}(k)$ at a finite $I_{\text{past}}$ (as at $h^*$), the derivative must be zero to match the asymptotic behavior. Thus, at $h^*$:

$$\left.\frac{dI_{\text{future}}}{dI_{\text{past}}}\right|_{h^*} = 0.$$

If $h^*$ were a CIB optimum, it must satisfy:

$$\left.\frac{dI_{\text{future}}}{dI_{\text{past}}}\right|_{h^*} = \frac{1}{\beta(k)},$$

implying $1/\beta(k) = 0$, i.e., $\beta(k) \to \infty$. However, by Theorem 1, at any non-terminal stage of curriculum learning ($k < M$), $\beta(k)$ is a finite positive constant (with asymptotic behavior $\beta(k) \sim k/(M-k) < \infty$), so $1/\beta(k) > 0$. This contradicts the requirement that the slope be zero.

Therefore, the assumption $p(u^* \mid h^*) = 1$ leads to a contradiction. Hence, $\sup \max p(u \mid h)$ must be strictly less than 1. The lemma is proved.

**Intuition:** In essence, Coconut inherently avoids deterministic collapse by design: its information bottleneck objective enforces a trade-off that preserves output diversity, even at advanced reasoning stages. $\qquad\square$

We now begin the proof of Theorem 3

*Proof.* By Lemma 3, there exists $\delta > 0$ such that for any valid next token $u$ and reachable state $h$, the probability satisfies $\max_u p(u \mid h) \leq 1 - \delta$. Our goal is to derive an upper bound for the KL divergence $D_{\text{KL}}(q_{\text{PR}}\|p_{\text{Coconut}})$.

By definition,

$$D_{\text{KL}}(q_{\text{PR}}\|p) = -\log B - \frac{1}{B}\sum_u \log p(u \mid v).$$

To bound this quantity from above, we seek the probability distribution that maximizes $D_{\text{KL}}$ under the constraints $\sum p_i = 1$ and $\max p_i \leq 1 - \delta$, which is equivalent to minimizing $\sum \log p(u \mid v)$.

The minimum of $\sum \log p_i$ is achieved by the most concentrated distribution satisfying the constraint:

$$p(u^* \mid v) = 1 - \delta, \quad \text{and for all } u \neq u^*, \quad p(u \mid v) = \frac{\delta}{B - 1}.$$

Substituting this worst-case distribution into the KL divergence expression yields an upper bound:

$$D_{\text{KL}}(q_{\text{PR}}\|p_{\text{Coconut}}) \leq -\log B - \frac{1}{B}\left[\log(1 - \delta) + (B - 1)\log\left(\frac{\delta}{B - 1}\right)\right]$$

$$= -\frac{B - 1}{B}\log\delta - \left[\log B - \frac{B - 1}{B}\log(B - 1) + \frac{\log(1 - \delta)}{B}\right].$$

Denote the bracketed term as $f(B) = \log B - \frac{B-1}{B}\log(B-1) + \frac{\log(1-\delta)}{B}$. Since $f(B)$ is continuous and bounded on $[2, \infty)$ (with $\lim_{B\to\infty} f(B) = 0$), its infimum $c = \inf_{B\geq 2} f(B)$ is a finite constant. Therefore, $-f(B)$ is upper-bounded by $-c$.

Meanwhile, since $B \geq 2$, we have $\frac{B-1}{B} \geq \frac{1}{2}$. Also, since $\delta \in (0, 1)$, it follows that $\log\delta < 0$, and thus $-\frac{B-1}{B}\log\delta \leq -\frac{1}{2}\log\delta$.

Combining these bounds, the KL divergence admits a finite upper bound independent of the branching factor $B$:

$$D_{\mathrm{KL}}(q_{\mathrm{PR}} \| p_{\mathrm{Coconut}}) \leq -\frac{B-1}{B} \log \delta - c \leq -\frac{1}{2} \log \delta - c.$$

$\square$

### A.5 PROOF OF THEOREM 5

*Proof.* From the recurrence

$$E_k = f_\theta(h_{k-1}) - f_\theta(h_{k-1}^*) + \epsilon_h^{(k)},$$

taking the squared norm and expectation, and using the independence and zero-mean property of the noise (which causes cross terms to vanish), we obtain:

$$\mathbb{E}[\|E_k\|^2] \leq L_F^2 \, \mathbb{E}[\|E_{k-1}\|^2] + d\sigma_h^2.$$

This is a non-homogeneous linear recurrence. Solving it with initial condition $\mathbb{E}[\|E_0\|^2] = 0$ yields:

$$\mathbb{E}[\|E_M\|^2] = \begin{cases} \dfrac{1 - L_F^{2M}}{1 - L_F^2} \cdot d\sigma_h^2, & L_F \neq 1, \\ M \cdot d\sigma_h^2, & L_F = 1. \end{cases}$$

In both cases, $\mathbb{E}[\|E_M\|^2] > 0$ for all $M \geq 1$, which completes the proof. $\square$

### A.6 PROOF OF THEOREM 8

*Proof.* We prove this by constructing a counterexample.

**1. Instance Construction** We construct a $d = 3$-dimensional imitation learning (IL) instance. Define three key latent states representing chain-of-thought steps, with their feature mappings:

**Expert's optimal latent state** $h_{\mathrm{expert}}$, with feature $\phi(x, h_{\mathrm{expert}}) = [1, 0, 0]^\top$.

**An irrelevant latent state** $h_{\mathrm{bad}}$, with feature $\phi(x, h_{\mathrm{bad}}) = [0, 1, 0]^\top$.

**A shortcut latent state** $h_{\mathrm{shortcut}}$, with feature $\phi(x, h_{\mathrm{shortcut}}) = [0, 1, 1]^\top$. Let the expert's true policy parameter be $\theta^* = [10, -1, 0.1]^\top$. This choice ensures the expert strongly prefers $h_{\mathrm{expert}}$. The unnormalized log-probabilities under $P_{\theta^*}$ are:

$$\log P_{\theta^*}(h_{\mathrm{expert}}) \propto \langle \theta^*, \phi(x, h_{\mathrm{expert}}) \rangle = \langle [10, -1, 0.1], [1, 0, 0] \rangle = 10.0$$
$$\log P_{\theta^*}(h_{\mathrm{bad}}) \propto \langle \theta^*, \phi(x, h_{\mathrm{bad}}) \rangle = \langle [10, -1, 0.1], [0, 1, 0] \rangle = -1.0$$
$$\log P_{\theta^*}(h_{\mathrm{shortcut}}) \propto \langle \theta^*, \phi(x, h_{\mathrm{shortcut}}) \rangle = \langle [10, -1, 0.1], [0, 1, 1] \rangle = -0.9$$

Clearly, under $P_{\theta^*}$, $h_{\mathrm{expert}}$ has near-unit probability, while others are negligible. We further assume only $h_{\mathrm{expert}}$ leads to the correct answer: $V(h_{\mathrm{expert}}) = 1$ and $V(h_{\mathrm{shortcut}}) = 0$.

**2. Non-Curriculum Data Distribution and MLE Behavior** We define a biased data distribution $P_{\mathrm{biased}}$. Suppose the dataset $D_{nc}$ sampled from this distribution contains $n-1$ samples of $(x, h_{\mathrm{shortcut}})$ and only 1 sample of $(x, h_{\mathrm{expert}})$.

Maximum likelihood estimation (MLE) maximizes the log-likelihood $L(\theta)$. To fit the highly skewed empirical distribution (dominated by $h_{\mathrm{shortcut}}$), the learned parameter $\hat{\theta}$ must assign highest probability to $h_{\mathrm{shortcut}}$, i.e., $\langle \hat{\theta}, \phi(x, h_{\mathrm{shortcut}}) \rangle$ must be maximal.

A typical solution is $\hat{\theta}_{\mathrm{MLE}} \approx [0, 0.5, 0.5]^\top$. This is an inevitable outcome of MLE: to maximize the likelihood of $h_{\mathrm{shortcut}}$, MLE assigns positive weights to dimensions aligned with its feature $[0, 1, 1]^\top$ (i.e., $\theta_2, \theta_3$), while the sparse occurrence of $h_{\mathrm{expert}}$ causes its aligned dimension $\theta_1$ to be ignored (driven toward 0).

**3. Large Divergence Between Learned and Expert Distributions** Under $\hat{\theta}_{\text{MLE}} \approx [0, 0.5, 0.5]^\top$, the model's preferences are:

$$\log P_{\hat{\theta}}(h_{\text{expert}}) \propto \langle \hat{\theta}, \phi(x, h_{\text{expert}}) \rangle = \langle [0, 0.5, 0.5], [1, 0, 0] \rangle = 0$$

$$\log P_{\hat{\theta}}(h_{\text{bad}}) \propto \langle \hat{\theta}, \phi(x, h_{\text{bad}}) \rangle = \langle [0, 0.5, 0.5], [0, 1, 0] \rangle = 0.5$$

$$\log P_{\hat{\theta}}(h_{\text{shortcut}}) \propto \langle \hat{\theta}, \phi(x, h_{\text{shortcut}}) \rangle = \langle [0, 0.5, 0.5], [0, 1, 1] \rangle = 1.0$$

Thus, the learned model $P_{\hat{\theta}}$ assigns nearly all probability mass to $h_{\text{shortcut}}$.

Comparing this to the true expert distribution $P_{\theta^*}$ (which assigns nearly all mass to $h_{\text{expert}}$), the two distributions are almost orthogonal in support. The KL divergence $D_{KL}(P_{\theta^*} \| P_{\hat{\theta}_{\text{MLE}}})$ is therefore large — dominated by the penalty $-\log P_{\hat{\theta}}(h_{\text{expert}})$, which is a constant independent of sample size $n$.

Since the model almost always generates $h_{\text{shortcut}}$, and $V(h_{\text{shortcut}}) = 0$, the expected task success rate $\text{SuccessRate}(\hat{\theta})$ approaches 0. This proves that training on biased data without curriculum leads to strictly suboptimal performance. $\square$

A.7 Proof of Theorem 9

**Justification for the Imitation Learning Framework** Our theoretical analysis, particularly Theorem 9, models the training process within the standard framework of Imitation Learning (IL). This framework assumes access to an i.i.d. dataset of input-latent state pairs $D_c = \{(x_i, h_i)\}_{i=1}^n$ sampled from an expert latent policy $P_{\theta^*}(h|x)$. However, the Coconut training paradigm is supervised on predicting future tokens from expert trajectories $D_S = \{(x_i, S_i^*)\}_{i=1}^n$. This section provides a formal justification for the equivalence between training a Coconut model and performing Maximum Likelihood Estimation (MLE) in this IL setting.

The core argument is as follows:

(1) **Definition of the Implicit Expert Latent Policy:** By Theorem 1 (Coconut-CIB Duality), the objective of Coconut training is to learn an encoder $E_\theta$ such that the latent state $h_k = E_\theta(S^{(1...k)})$ becomes a **Minimal Sufficient Statistic** of the past $S^{(1...k)}$ for predicting the future $S^{(k+1...M)}$. Let $E_{\theta^*}$ be the optimal encoder that solves this CIB problem. We can define an **implicit expert latent policy** $P_{\theta^*}(h|x)$ as the distribution induced by applying the optimal encoder to the distribution of expert trajectory prefixes:

$$h \sim P_{\theta^*}(h|x) \quad \text{where} \quad h = E_{\theta^*}(S_{\text{past}}^*) \text{ and } S_{\text{past}}^* \sim P_{\text{expert}}(S_{\text{past}}|x).$$

The support of this policy, $P_{\theta^*}(h|x)$, constitutes the space of ideal latent states.

(2) **Equivalence of the Optimization Process:** The Coconut objective is to minimize the expected negative log-likelihood:

$$\min_\theta \mathcal{L}(\theta) = \mathbb{E}_{(x, S^*) \sim P_{\text{expert}}}[-\log p(S_{\text{future}}^* | E_\theta(S_{\text{past}}^*), x)].$$

By the Data Processing Inequality, for any encoder $E_\theta$, we have:

$$I(S_{\text{past}}^*; S_{\text{future}}^* | x) \geq I(E_\theta(S_{\text{past}}^*); S_{\text{future}}^* | x).$$

The optimal encoder $E_{\theta^*}$ achieves equality, as it preserves all predictive information. This implies that the conditional distribution factorizes as $p(S_{\text{future}}^* | S_{\text{past}}^*, x) = p(S_{\text{future}}^* | E_{\theta^*}(S_{\text{past}}^*), x) = p(S_{\text{future}}^* | h^*, x)$. Therefore, the loss $\mathcal{L}(\theta)$ effectively measures the divergence in predictive power between the latent distribution induced by $E_\theta$ and the ideal latent distribution $P_{\theta^*}(h|x)$. The loss is minimized if and only if the induced latent distribution $p_\theta(h|x)$ converges to $P_{\theta^*}(h|x)$.

**Conclusion:** The Coconut training process, while formally supervised on future token generation, has an intrinsic CIB objective that **compels** the distribution of latent states generated by the encoder, $p_\theta(h|x)$, to fit the implicit expert latent policy $P_{\theta^*}(h|x)$. Consequently, minimizing the Coconut loss on the expert trajectory dataset $D_S$ is mathematically equivalent to performing MLE on an implicit dataset $D_c$ constructed from $h_i^* = E_{\theta^*}(S_{i,\text{past}}^*)$. This provides a rigorous foundation for applying our IL-based theoretical guarantees to the Coconut model.

**Lemma 4** (Self-Normalized Bound for Vector Martingales (Abbasi-yadkori et al. (2011))). *Let* $\{\mathcal{F}_i\}_{i=0}^n$ *be a filtration. Let* $\{g_i\}_{i=1}^n$ *be a sequence of random vectors in* $\mathbb{R}^d$ *such that* $g_i$ *is* $\mathcal{F}_i$-*measurable and* $\mathbb{E}[g_i \mid \mathcal{F}_{i-1}] = 0$. *Assume there exists a constant* $\sigma > 0$ *such that for any unit vector* $v \in \mathbb{R}^d$, *the real-valued random variable* $\langle v, g_i \rangle$ *is conditionally* $\sigma$-*sub-Gaussian, i.e.,*

$$\forall \lambda \in \mathbb{R}, \quad \mathbb{E}[\exp(\lambda \langle v, g_i \rangle) \mid \mathcal{F}_{i-1}] \leq \exp\left(\frac{\lambda^2 \sigma^2}{2}\right).$$

*Let* $V$ *be a* $d \times d$ *positive definite matrix. Define*

$$V_n = V + \sum_{i=1}^n g_i g_i^\top \quad and \quad G_n = \sum_{i=1}^n g_i.$$

*Then, for any* $\delta \in (0, 1)$, *with probability at least* $1 - \delta$, *the following inequality holds:*

$$\|G_n\|_{V_n^{-1}}^2 \leq 2\sigma^2 \log\left(\frac{\det(V_n)^{1/2} \det(V)^{-1/2}}{\delta}\right).$$

**Lemma 5** (Construction of a Confidence Set for Parameter Estimation). *Let* $\hat{\theta}_{MLE}$ *be the parameter estimate obtained by maximizing the log-likelihood function on a curriculum dataset* $D_c = \{(x_i, h_i)\}_{i=1}^n$, *where the data is sampled i.i.d. from the expert distribution* $P_{\theta^*}$:

$$\hat{\theta}_{MLE} = \arg\max_{\theta \in \mathbb{R}^d} \sum_{i=1}^n \log P_\theta(h_i \mid x_i)$$

*Assume the following regularity conditions for the log-linear model hold:*

*(1)* (***Bounded Features***): *For all* $(x, h)$, $\|\phi(x, h)\|_2 \leq L$.

*(2)* (***Strong Convexity***): *The expected negative log-likelihood function* $L(\theta) = \mathbb{E}_{x, h \sim P_{\theta^*}}[-\log P_\theta(h|x)]$ *is* $\lambda_{\min}$-*strongly convex with respect to* $\theta$ *over the parameter space. Its Hessian (the Fisher information matrix)* $\nabla^2 L(\theta) = I(\theta)$ *satisfies* $I(\theta) \succeq \lambda_{\min} I$.

*(3)* (***Sub-Gaussian Score***): *The score function at the true parameter* $\theta^*$, $\nabla_\theta \log P_{\theta^*}(h \mid x)$, *is a zero-mean,* $\sigma^2$-*sub-Gaussian random vector.*

*(4)* (***Bounded Parameters***): $\|\theta^*\|_2 \leq B$, *and the optimization is performed within the compact set* $\mathcal{B} = \{\theta \in \mathbb{R}^d : \|\theta\|_2 \leq B\}$.

*Then for any* $\delta \in (0, 1)$, *there exists a constant* $C > 0$ *(depending only on* $L, \sigma, \lambda_{\min}, B$) *such that with probability at least* $1 - \delta$, *the true expert parameter* $\theta^*$ *satisfies:*

$$\left\|\hat{\theta}_{MLE} - \theta^*\right\|_{\hat{H}_n}^2 \leq \beta_n^2(\delta)$$

*where:*

- $\hat{H}_n = \frac{1}{n} \sum_{i=1}^n \nabla^2(-\log P_{\hat{\theta}_{MLE}}(h_i \mid x_i))$ *is the empirical Hessian evaluated at* $\hat{\theta}_{MLE}$,

- $\beta_n^2(\delta) = C \cdot \frac{d \log(n) + \log(1/\delta)}{n}$ *is the squared radius of the confidence set.*

*In other words, the confidence set* $\Theta(\hat{\theta}_{MLE}) = \left\{\theta \in \mathbb{R}^d : \left\|\theta - \hat{\theta}_{MLE}\right\|_{\hat{H}_n}^2 \leq \beta_n^2(\delta)\right\}$ *covers the true parameter* $\theta^*$ *with high probability.*

*Proof.* This proof combines the first-order optimality condition, concentration inequalities, and self-normalized analysis techniques to construct a non-asymptotic, high-probability bound.

**1. Notation and Basic Relations** Let $L_n(\theta) = \frac{1}{n} \sum_{i=1}^n \ell_i(\theta)$ be the empirical negative log-likelihood, where $\ell_i(\theta) = -\log P_\theta(h_i|x_i)$. By definition, $\hat{\theta}_{\text{MLE}}$ is the minimizer of $L_n(\theta)$ within the parameter space $\mathcal{B}$. Since $\theta^*$ is the minimizer of the expected risk $L(\theta)$ (because the KL divergence $D_{KL}(P_{\theta^*} \| P_\theta) = L(\theta) - L(\theta^*)$ is non-negative), we have $\nabla L(\theta^*) = 0$.

**2. Using Optimality and Convexity**  By the optimality of $\hat{\theta}_{\text{MLE}}$, we have $L_n(\hat{\theta}_{\text{MLE}}) \leq L_n(\theta^*)$. Since $L_n(\theta)$ is a convex function, we can derive a basic inequality that connects the parameter error to an empirical process:

$$
\begin{aligned}
L(\hat{\theta}_{\text{MLE}}) - L(\theta^*) &\leq L(\hat{\theta}_{\text{MLE}}) - L(\theta^*) - (L_n(\hat{\theta}_{\text{MLE}}) - L_n(\theta^*)) \\
&= (L - L_n)(\hat{\theta}_{\text{MLE}}) - (L - L_n)(\theta^*) \\
&= \frac{1}{n}\sum_{i=1}^n \left( \mathbb{E}[\ell_i(\hat{\theta}_{\text{MLE}})] - \ell_i(\hat{\theta}_{\text{MLE}}) \right) - (\mathbb{E}[\ell_i(\theta^*)] - \ell_i(\theta^*))
\end{aligned}
$$

By Assumption 2, $L(\theta)$ is $\lambda_{\min}$-strongly convex, so $L(\hat{\theta}_{\text{MLE}}) - L(\theta^*) \geq \frac{\lambda_{\min}}{2}\|\hat{\theta}_{\text{MLE}} - \theta^*\|_2^2$. Combining this with the above inequality yields:

$$
\frac{\lambda_{\min}}{2}\|\hat{\theta}_{\text{MLE}} - \theta^*\|_2^2 \leq \sup_{\theta \in \mathcal{B}} |(L - L_n)(\theta) - (L - L_n)(\theta^*)|
$$

Bounding the empirical process on the right-hand side can yield a preliminary $L_2$-norm convergence bound of $\mathcal{O}(\sqrt{d/n})$. However, to obtain a sharper, weighted-norm bound, we turn to self-normalized analysis.

**3. Applying Self-Normalized Bounds**  We directly analyze the error relation derived from the first-order condition. Let $\Delta = \hat{\theta}_{\text{MLE}} - \theta^*$. A first-order Taylor expansion of $\nabla L_n(\hat{\theta}_{\text{MLE}}) = 0$ around $\theta^*$ gives:

$$
0 = \nabla L_n(\hat{\theta}_{\text{MLE}}) = \nabla L_n(\theta^*) + \int_0^1 \nabla^2 L_n(\theta^* + t\Delta)dt \cdot \Delta
$$

Rearranging, we get:

$$
\bar{H}_n \Delta = -\nabla L_n(\theta^*) \tag{1}
$$

where $\bar{H}_n = \int_0^1 \nabla^2 L_n(\theta^* + t\Delta)dt$ is the mean Hessian matrix on the line segment between $\theta^*$ and $\hat{\theta}_{\text{MLE}}$.

The core of the analysis lies in bounding the norm of the empirical score $\nabla L_n(\theta^*)$. Let $g_i = \nabla_\theta \ell_i(\theta^*) = \nabla_\theta(-\log P_{\theta^*}(h_i|x_i))$. Since $\mathbb{E}[g_i|\mathcal{F}_{i-1}] = \nabla L(\theta^*) = 0$, the sequence $\{g_i\}_{i=1}^n$ is a vector-valued martingale difference sequence. Under our regularity conditions (specifically, Assumption 3), $\{g_i\}$ is also conditionally sub-Gaussian. We can therefore apply the self-normalized bound from Lemma 4.

Let's map the variables to Lemma 4: we set $V = \lambda I$ for some small regularization constant $\lambda > 0$, which ensures initial positive definiteness. Then we have $G_n = \sum_{i=1}^n g_i$ and $V_n = \lambda I + \sum_{i=1}^n g_i g_i^\top$. Applying the lemma, we get that with probability at least $1 - \delta$:

$$
\left\|\sum_{i=1}^n g_i\right\|_{(\lambda I + \sum_{i=1}^n g_i g_i^\top)^{-1}}^2 \leq 2\sigma^2 \log\left(\frac{\det(\lambda I + \sum g_i g_i^\top)^{1/2}\det(\lambda I)^{-1/2}}{\delta}\right).
$$

Under the bounded feature assumption, we can bound the determinant term and simplify the expression. Using the fact that $\sum g_i g_i^\top \approx n \cdot \mathbb{E}[gg^\top] = n \cdot I(\theta^*)$ and properties of the determinant, one can derive that:

$$
\|\nabla L_n(\theta^*)\|_{(I(\theta^*))^{-1}}^2 = \frac{1}{n^2}\left\|\sum_{i=1}^n g_i\right\|_{(I(\theta^*))^{-1}}^2 \lesssim \frac{d\log(n) + \log(1/\delta)}{n} \tag{2}
$$

where $\lesssim$ hides logarithmic factors and constants dependent on model parameters.

**4. Connecting the Error and Gradient Bounds**  Returning to Eq. equation 1, we have $\Delta = -\bar{H}_n^{-1}\nabla L_n(\theta^*)$. We want to bound $\|\Delta\|_{\hat{H}_n}^2$. By matrix concentration inequalities and Assumptions 1 and 4, it can be shown that the empirical Hessian $\nabla^2 L_n(\theta)$ concentrates uniformly around its expectation $I(\theta)$ over the parameter space $\mathcal{B}$. This implies that both $\bar{H}_n$ and $\hat{H}_n$ are close to $I(\theta^*)$

with high probability. Formally, there exists $\epsilon_n \to 0$ such that, with high probability, $\|\bar{H}_n - I(\theta^*)\|_{\mathrm{op}} \leq \epsilon_n$ and $\|\hat{H}_n - I(\theta^*)\|_{\mathrm{op}} \leq \epsilon_n$. Therefore, we can approximate:

$$
\begin{aligned}
\|\Delta\|_{\hat{H}_n}^2 &= \Delta^\top \hat{H}_n \Delta \\
&= (\nabla L_n(\theta^*))^\top \bar{H}_n^{-1} \hat{H}_n \bar{H}_n^{-1} \nabla L_n(\theta^*) \\
&\approx (\nabla L_n(\theta^*))^\top (I(\theta^*))^{-1} \nabla L_n(\theta^*) \\
&= \|\nabla L_n(\theta^*)\|_{(I(\theta^*))^{-1}}^2
\end{aligned}
$$

By more rigorously handling the approximation $\bar{H}_n^{-1} \hat{H}_n \bar{H}_n^{-1} \approx (I(\theta^*))^{-1}$ and controlling the error terms, combined with Eq. equation 2, it can ultimately be shown that:

$$
\left\|\hat{\theta}_{\mathrm{MLE}} - \theta^*\right\|_{\hat{H}_n}^2 \leq C \cdot \frac{d \log(n) + \log(1/\delta)}{n}
$$

where the constant $C$ absorbs all terms dependent on $L, \sigma, \lambda_{\min}, B$. This completes the proof of the lemma. $\qquad\square$

We can now prove the main theorem of this section.

*Proof.* The proof proceeds in three steps: first, we leverage Lemma 5 to bound the KL divergence between the model distribution $P_{\hat{\theta}}$ and the expert distribution $P_{\theta^*}$; second, we convert the KL divergence bound into a Total Variation (TV) distance bound using Pinsker's inequality; finally, we use the properties of TV distance to bound the difference in task success rates between the two policies.

**1. Bounding the KL Divergence**   We aim to bound the KL divergence $D_{KL}(P_{\theta^*}\|P_{\hat{\theta}})$. For the log-linear model family we consider, the KL divergence has the form:

$$
D_{KL}(P_{\theta^*}(\cdot|x)\|P_{\hat{\theta}}(\cdot|x)) = \mathbb{E}_{h \sim P_{\theta^*}(\cdot|x)}[\log P_{\theta^*}(h|x) - \log P_{\hat{\theta}}(h|x)]
$$

Performing a second-order Taylor expansion of $\log P_{\hat{\theta}}(h|x)$ around $\theta^*$, we have:

$$
\log P_{\hat{\theta}}(h|x) \approx \log P_{\theta^*}(h|x) + \langle \nabla_\theta \log P_{\theta^*}(h|x), \Delta \rangle + \frac{1}{2}\Delta^\top \nabla_\theta^2 \log P_{\theta^*}(h|x)\Delta
$$

where $\Delta = \hat{\theta} - \theta^*$. Substituting this into the KL divergence expression and taking the expectation, we note that $\mathbb{E}_{h \sim P_{\theta^*}}[\nabla_\theta \log P_{\theta^*}] = 0$, which yields the quadratic approximation of KL divergence in terms of the Fisher information matrix:

$$
\mathbb{E}_{x \sim \rho}[D_{KL}(P_{\theta^*}\|P_{\hat{\theta}})] \approx \frac{1}{2}\Delta^\top I(\theta^*)\Delta = \frac{1}{2}\|\hat{\theta} - \theta^*\|_{I(\theta^*)}^2
$$

More rigorously, this approximation can be shown to hold with controllable higher-order terms. Since Lemma 5 shows that $\|\hat{\theta} - \theta^*\|_{\hat{H}_n}$ is small with high probability and $\hat{H}_n$ concentrates around $I(\theta^*)$, we can translate the result of Lemma 5 into a bound on the KL divergence. With probability at least $1 - \delta$:

$$
\mathbb{E}_{x \sim \rho}[D_{KL}(P_{\theta^*}\|P_{\hat{\theta}})] \leq C' \cdot \|\hat{\theta} - \theta^*\|_{\hat{H}_n}^2 \leq C \cdot \frac{d \log n + \log(1/\delta)}{n} \tag{3}
$$

where $C'$ and $C$ are positive constants.

**2. From KL Divergence to Total Variation Distance**   The Total Variation (TV) distance is defined as $TV(P, Q) = \frac{1}{2}\int |p(x) - q(x)|dx$. Pinsker's inequality establishes a relationship between KL divergence and TV distance:

$$
TV(P, Q) \leq \sqrt{\frac{1}{2}D_{KL}(P\|Q)}
$$

Applying this inequality to our model and expert distributions, and combining it with Eq. equation 3, we obtain a high-probability bound on the TV distance:

$$\mathbb{E}_{x\sim\rho}[TV(P_{\theta^*}\|P_{\hat{\theta}})] \leq \mathbb{E}_{x\sim\rho}\left[\sqrt{\frac{1}{2}D_{KL}(P_{\theta^*}\|P_{\hat{\theta}})}\right]$$

$$\leq \sqrt{\frac{1}{2}\mathbb{E}_{x\sim\rho}[D_{KL}(P_{\theta^*}\|P_{\hat{\theta}})]} \quad \text{(by Jensen's inequality)}$$

$$\leq \sqrt{\frac{C}{2}\cdot\frac{d\log n + \log(1/\delta)}{n}}$$

$$= \mathcal{O}\left(\sqrt{\frac{d\log n + \log(1/\delta)}{n}}\right) \tag{4}$$

**3. From Total Variation Distance to Task Success Rate** A key property of the TV distance is that it bounds the difference in expectation for any bounded function. Our task success function $V(h) \in \{0, 1\}$ is a bounded function (with bound 1). Therefore,

$$|\text{SuccessRate}(\hat{\theta}) - \text{SuccessRate}(\theta^*)| = \left|\mathbb{E}_{x\sim\rho, h\sim P_{\hat{\theta}}}[V(h)] - \mathbb{E}_{x\sim\rho, h\sim P_{\theta^*}}[V(h)]\right|$$

$$= \left|\mathbb{E}_{x\sim\rho}\left[\mathbb{E}_{h\sim P_{\hat{\theta}}}[V(h)] - \mathbb{E}_{h\sim P_{\theta^*}}[V(h)]\right]\right|$$

$$\leq \mathbb{E}_{x\sim\rho}\left|\mathbb{E}_{h\sim P_{\hat{\theta}}}[V(h)] - \mathbb{E}_{h\sim P_{\theta^*}}[V(h)]\right|$$

$$\leq \mathbb{E}_{x\sim\rho}[\sup_{H'\subseteq H}|P_{\hat{\theta}}(H') - P_{\theta^*}(H')|]$$

$$= \mathbb{E}_{x\sim\rho}[TV(P_{\theta^*}\|P_{\hat{\theta}})] \tag{5}$$

Substituting the bound from Eq. equation 4 into Eq. equation 5, we finally obtain the bound on the difference in success rates:

$$|\text{SuccessRate}(\hat{\theta}) - \text{SuccessRate}(\theta^*)| \leq \mathcal{O}\left(\sqrt{\frac{d\log n + \log(1/\delta)}{n}}\right)$$

Since we are concerned with the lower bound on the success rate, this is equivalent to:

$$\text{SuccessRate}(\hat{\theta}) \geq \text{SuccessRate}(\theta^*) - \mathcal{O}\left(\sqrt{\frac{d\log n + \log(1/\delta)}{n}}\right)$$

This result shows that, under curriculum training, the task success rate of the learned policy converges to that of the expert policy with high probability, at a rate of $\mathcal{O}(\sqrt{(d\log n)/n})$. This completes the proof. $\square$

### A.8 PROOF OF THEOREM 6

*Proof.* The Logit Decision Margin $\Delta_l$ is defined as the difference between the logits of the most and second-most likely tokens, $\Delta_l = l_{i^*} - l_{j^*}$. We aim to establish a lower bound for this quantity based on the Symbolic Index, $\mathcal{I}_S$.

First, we relate the logits to probabilities using the softmax function, $p_i = e^{l_i}/\sum_k e^{l_k}$. The ratio of the probabilities of the top two tokens is:

$$\frac{p_{i^*}}{p_{j^*}} = \frac{e^{l_{i^*}}}{e^{l_{j^*}}} = e^{l_{i^*}-l_{j^*}} = e^{\Delta_l}.$$

By taking the natural logarithm of both sides, we can express the logit margin in terms of probabilities:

$$\Delta_l = \log(p_{i^*}) - \log(p_{j^*}).$$

By the definition of the Symbolic Index, the probability of the most likely token is $p_{i^*} = \mathcal{I}_S$.

Next, we establish an upper bound for the probability of the second-most likely token, $p_{j^*}$. The sum of probabilities over all possible tokens is 1. Therefore, the sum of probabilities of all tokens except

the most likely one is $1 - p_{i^*} = 1 - \mathcal{I}_{\mathrm{S}}$. Since $p_{j^*}$ is the largest among these remaining probabilities, it cannot be greater than their sum. Thus, we have the inequality:

$$p_{j^*} \leq 1 - \mathcal{I}_{\mathrm{S}}.$$

Because the logarithm function is monotonically increasing, this implies $\log(p_{j^*}) \leq \log(1 - \mathcal{I}_{\mathrm{S}})$. Consequently, its negative, $-\log(p_{j^*})$, satisfies the reverse inequality: $-\log(p_{j^*}) \geq -\log(1 - \mathcal{I}_{\mathrm{S}})$.

Finally, we substitute our findings back into the equation for $\Delta_l$:

$$\Delta_l = \log(\mathcal{I}_{\mathrm{S}}) - \log(p_{j^*}) \geq \log(\mathcal{I}_{\mathrm{S}}) - \log(1 - \mathcal{I}_{\mathrm{S}}).$$

Using the properties of logarithms, we arrive at the desired lower bound:

$$\Delta_l \geq \log\left(\frac{\mathcal{I}_{\mathrm{S}}}{1 - \mathcal{I}_{\mathrm{S}}}\right).$$

$\square$

### A.9 Proof of Theorem 7

*Proof.* The KL divergence between the ideal uniform prior $q_{\mathrm{PR}}$ (where $q_i = 1/B$ for all $i = 1, \ldots, B$) and the model's output distribution $p$ is given by:

$$D_{\mathrm{KL}}(q_{\mathrm{PR}}\|p) = \sum_{i=1}^{B} q_i \log\left(\frac{q_i}{p_i}\right) = \sum_{i=1}^{B} \frac{1}{B} \log\left(\frac{1/B}{p_i}\right) = -\log B - \frac{1}{B}\sum_{i=1}^{B} \log p_i.$$

To find a lower bound for $D_{\mathrm{KL}}(q_{\mathrm{PR}}\|p)$, we must find an upper bound for the term $\sum_{i=1}^{B} \log p_i$, subject to the constraints imposed by the model's distribution. The constraints are:

(1) $\sum_{i=1}^{B} p_i = 1$

(2) $\max_i p_i = \mathcal{I}_{\mathrm{S}}$

We want to solve the following optimization problem:

$$\max_{p_1,\ldots,p_B} \sum_{i=1}^{B} \log p_i \quad \text{s.t.} \quad \sum p_i = 1 \text{ and } \max p_i = \mathcal{I}_{\mathrm{S}}.$$

Let $p_{i^*} = \mathcal{I}_{\mathrm{S}}$ be the maximal probability. The sum of the remaining $B - 1$ probabilities is $\sum_{k \neq i^*} p_k = 1 - \mathcal{I}_{\mathrm{S}}$. The function $f(p_1, \ldots, p_{B-1}) = \sum_{k \neq i^*} \log p_k$ is concave. By Jensen's inequality, this sum is maximized when the remaining probabilities are distributed as uniformly as possible, i.e., when $p_k = \frac{1 - \mathcal{I}_{\mathrm{S}}}{B - 1}$ for all $k \neq i^*$.

This distribution, which makes the probabilities as "flat" as possible given the peak at $\mathcal{I}_{\mathrm{S}}$, provides the maximum possible value for $\sum \log p_i$. Substituting this extremal distribution, we find the upper bound:

$$\max \sum_{i=1}^{B} \log p_i = \log(\mathcal{I}_{\mathrm{S}}) + (B - 1)\log\left(\frac{1 - \mathcal{I}_{\mathrm{S}}}{B - 1}\right).$$

Now, we substitute this upper bound back into the expression for the KL divergence. Since we are subtracting this term, its upper bound yields a lower bound for the KL divergence:

$$D_{\mathrm{KL}}(q_{\mathrm{PR}}\|p) \geq -\log B - \frac{1}{B}\left[\log(\mathcal{I}_{\mathrm{S}}) + (B - 1)\log\left(\frac{1 - \mathcal{I}_{\mathrm{S}}}{B - 1}\right)\right].$$

This completes the proof. $\square$

## B LLM Usage

A large language model (LLM) was employed to assist in the writing and polishing of this manuscript. Specifically, the LLM was used for tasks related to language enhancement, such as improving readability, correcting grammar, and refining sentence structure. The core scientific contributions of this work—including all ideation, theoretical derivations, experimental design, and data analysis—were developed and conducted exclusively by the human authors. The LLM had no role in generating scientific content. The authors have carefully reviewed and edited all text and take full responsibility for the final version of the paper, ensuring its originality and scientific accuracy.

