# OpenReview forum: "The Theoretical Benefits and Limitations of Latent Chain-of-Thought Reasoning"
_ICLR.cc/2026/Conference — Submitted to ICLR 2026_

### Official Review · Reviewer_wX92 · 2025-10-23

**Soundness:** 3
**Presentation:** 2
**Contribution:** 2
**Rating:** 6
**Confidence:** 3

**Summary:**

This paper offers a theoretical framework to explain the performance discrepancy between explicit CoT reasoning and latent CoT reasoning. The authors introduce the concept of a Symbolic Index to quantify decisional certainty and unify the two paradigms under an exploration–execution trade-off.

**Strengths:**

1. Originality is sufficient. The introduction of the Symbolic Index is elegant and insightful. It connects decisional certainty, noise robustness, and exploration capacity under one quantitative measure.

2. The paper derives a series of results (Coconut–CIB duality, symbolic stability bound, KL-divergence trade-off) that make intuitive sense and are consistent with observed empirical phenomena.

**Weaknesses:**

1. In Sections 4.1-4.3, the authors theoretically prove, through the lens of the information bottleneck, that explicit CoT is stable but lacks exploration, whereas latent CoT exhibits the opposite behavior. However, I don’t find it particularly novel, as this is already a well-known fact. The authors merely examine it from a different perspective without providing any new insights.

2. The interesting point, in my view, lies in Section 4.4, where the authors leverage the symbolic index to unify the exploration–execution trade-off; however, the work lacks crucial experiments to verify that employing this index can effectively lead to a well-trained model.

3. The authors have placed all three main contributions into Section 4, which makes the section feel overcrowded and compact, and the overall logic is not very smooth. It would be better to separate them and use more straightforward explanations to connect the ideas. In the experimental part, the designed experiments do not seem to effectively validate the claims made in Section 4.4.

4. Strong assumptions in Theorem 1; Distribution-mismatch theorems (8–9) are too abstract.

**Questions:**

No more.

---

> ### Author Response · Authors · 2025-11-21
>
> **Q1**: In Sections 4.1-4.3, the authors theoretically prove, through the lens of the information bottleneck, that explicit CoT is stable but lacks exploration, whereas latent CoT exhibits the opposite behavior. However, I don’t find it particularly novel, as this is already a well-known fact. The authors merely examine it from a different perspective without providing any new insights.
>
> **A1**: We appreciate the reviewer’s comment. We fully agree that the *empirical* trade-off—that explicit CoT excels at precision while Latent CoT excels at exploration—is observed in prior works (e.g., Hao et al., 2024). However, we respectfully clarify that our contribution is not merely re-stating this phenomenon, but shifting the understanding from **empirical observation** (knowing *what* happens) to **theoretical causality** (proving *why* and *how* it essentially must happen).
>
> While the high-level behaviors are known, the underlying mathematical mechanisms governing them remained fundamentally unexplained. Our work provides the first rigorous derivation of these behaviors from first principles, offering specific new insights that go beyond the "well-known facts":
>
> *  **Mechanism of Failure:** Prior work observed *that* Latent CoT fails on math tasks, but not *why*. We introduce the concept of **sub-decisional perturbations** and prove a novel distinction: Explicit CoT possesses an inherent "reset mechanism" via discretization that filters noise (Theorem 4), whereas Latent CoT suffers from mathematically quantifiable **compounding error** in continuous space (Theorem 5). This proves that Latent CoT's failure on calculation is not merely a training issue, but a structural inevitability of continuous representations without discrete bottlenecks.
>
> *  **Theoretical Guarantee of Exploration:** It is not obvious that Latent CoT *must* explore better; empirically, it could have collapsed to a deterministic mode. We prove in Theorem 3 that due to the duality with the Information Bottleneck (Theorem 1), the model is mathematically forced into a regime with a **non-degenerate output distribution**. This provides a theoretical guarantee for exploration capabilities that empirical observation alone cannot supply.
>
> *  **Unification via the Symbolic Index:** We do not just analyze the two paradigms separately; we unify them under a single theoretical framework governed by the **Symbolic Index ($I_S$)**. We prove that decisional certainty ($I_S$) is the causal variable regulating the trade-off between the Logit Decision Margin (robustness) and the KL Divergence from uniform priors (exploration). This offers a new, quantifiable metric for designing future adaptive architectures, which is a significant step beyond qualitative descriptions of "stability vs. exploration."

---

> ### Author Response · Authors · 2025-11-21
>
> **Q2**: The interesting point, in my view, lies in Section 4.4, where the authors leverage the symbolic index to unify the exploration–execution trade-off; however, the work lacks crucial experiments to verify that employing this index can effectively lead to a well-trained model.
>
>
>
> **A2**: We thank the reviewer for recognizing the value of Section 4.4 and the Symbolic Index ($I_S$) as a unifying framework. We appreciate the suggestion to demonstrate how this index can be leveraged to train better models.
>
> While we acknowledge that a fully adaptive system that explicitly optimizes $I_S$ is an exciting direction for future work, we respectfully argue that our paper **already contains experimental verification** that regulating this index is necessary for a well-trained model—specifically through the **Curriculum Learning ablation study** presented in Table 1 and analyzed in Section 4.5.
>
> Here is how our existing experiments verify the necessity of managing $I_S$:
>
> * The Coconut curriculum (Hao et al., 2024) is not just a heuristic; it is effectively a schedule for managing the Symbolic Index.
>     *   **Stage $k=0$ (Pure CoT):** The model is forced into a High $I_S$ regime (discrete tokens). This establishes the "grounding" and high decision margin required for logical consistency.
>     *   **Stage $k \to M$ (Latent CoT):** The model gradually relaxes into a Low $I_S$ regime (continuous vectors). This enables the exploration capabilities we proved in Theorem 3.
>
> * The reviewer asks for evidence that employing this index (or rather, the principle behind it) leads to a well-trained model. Table 1 provides the counter-factual evidence:
>     *   **With Curriculum:** The model achieves 99.8% on ProntoQA.
>     *   **Without Curriculum:** The model collapses to 52.4%.
>     This catastrophic drop empirically confirms our theoretical claim: one cannot simply train a Latent CoT (Low $I_S$) from scratch because, as proved in Theorem 8, the distributional mismatch is insurmountable.
>
> * We position the Symbolic Index as a **diagnostic metric** and a **design principle** rather than a direct loss term in this specific work. It explains *when and why* current methods fail (compounding error in low $I_S$) or succeed (robustness in high $I_S$). It suggests that next-generation architectures should not be static (fixed CoT or fixed Latent) but dynamic. While building a system that actively monitors $I_S$ to switch modes during inference is a promising engineering challenge, our goal was to provide the theoretical foundation that proves *why* such a system would be beneficial.
>
> In summary, our work theoretically characterizes the trade-off and experimentally proves (via the failure of non-curriculum training) that managing decisional certainty is a prerequisite for convergence. We believe these insights provide the blueprint for the next stage of model design, even if building the adaptive model itself is future work.
>
> ---
> **Q3**:
>
> **A3**: We thank the reviewer for this constructive feedback on the manuscript structure.
>
> Upon reflection, we agree that Section 4 packs two distinct logical streams:
> * Test-time Analysis (4.1-4.4): Analyzing the intrinsic properties of the trained model (Exploration vs. Execution, Symbolic Index).
> * Training Dynamics (4.5): Proving the necessity of Curriculum Learning.
>
> Grouping them together may indeed feel disjointed, as 4.1-4.4 focus on "test-time behavior" while 4.5 focuses on "training stability."
>
> Action Plan for Revision:
> We will restructure the paper to improve flow and readability:
> * Split Section 4: We will keep Sections 4.1–4.4 as the core "Theoretical Analysis of Reasoning Mechanisms," culminating in the unification via the Symbolic Index.
> * New Section 5: We will promote Section 4.5 (Curriculum Learning) to a standalone section: "Theoretical Foundations of Training Dynamics." This separation clearly delineates the analysis of the inference process from the analysis of the learning process.
> * Better Signposting: We will add transitional text clarifying that while the first part explains why Latent CoT behaves differently, the second part explains why it must be trained in a specific way to achieve that behavior

---

> ### Author Response · Authors · 2025-11-21
>
> **Q4**: Strong assumptions in Theorem 1; Distribution-mismatch theorems (8–9) are too abstract.
>
> **A4**:
> We thank the reviewer for the insightful critique. We acknowledge that Section 4.5 was dense, and crucially, we realized that while we used SuccessRate in Theorem 9, we omitted its formal definition in the main text, which likely contributed to the feeling of "abstractness" and disconnection.
>
> Below, we clarify Theorem 1 and provide the missing definition that bridges our theoretical bounds to concrete task performance.
>
> 1. Clarifying Theorem 1
>
> We wish to clarify that Theorem 1 does not make strong assumptions about the model's capability (i.e., we do not assume the model is perfect). Instead, it is a structural analysis of the optimization objective implied by the Coconut architecture.
>
> - Standard IB Setup: The duality simply proves that the architectural constraint (compressing past tokens into $h_k$) combined with the prediction loss is mathematically equivalent to the Information Bottleneck (IB) objective (Tishby et al., 2000).
> - Implication: This is not a performance guarantee, but a mechanistic explanation. It reveals that Latent CoT is *forced* by its objective to trade off between retaining history (Exploration cost) and predicting the future (Execution precision). This justifies why we introduced the Symbolic Index as the measurable proxy for this trade-off.
>
> 2. Concretizing Theorems 8 & 9
>
> We agree that presenting Theorem 8 purely in terms of KL divergence ($D_{KL}$) made the practical consequences abstract. The missing link was a formal definition connecting distributional mismatch to task failure.
>
> Added Definition: Task Success Rate
>
> In the revision, we will explicitly define the Task Success Rate within our Imitation Learning framework before stating the theorems:
>
> Definition (Task Success Rate). Let $V(h, x) \in \{0, 1\}$ be a validity function where $V(h, x)=$ if the latent reasoning path $h$ decodes to the correct answer for input $x$. The Success Rate of a policy $\pi_\theta$ is its expected validity:
>
>  $\text{SuccessRate}(\theta) = E_{x \sim \mathcal{D}} \left[ E_{h \sim \pi_\theta(\cdot|x)} [V(h, x)] \right] $
>
> Reformulating the Theorems
>
> With this definition, we can now rigorously translate our "abstract" KL bounds into concrete performance gaps using Total Variation (TV) distance and Pinsker’s Inequality ($TV(P, Q) \le \sqrt{\frac{1}{2}D_{KL}(P||Q)}$).
>
> - Theorem 8 (No Curriculum $\to$ Failure):
>
> Originally, we showed the learned distribution $P_{\hat{\theta}}$ is orthogonal to the expert $P$  ($D_{KL}$ is large).
>
> Revised Interpretation: Because the distributions are disjoint (shortcut vs. reasoning), the TV distance is maximal. Consequently, the Success Rate Gap is **strictly bounded away from 0**. The model provably fails to learn the reasoning policy.
>
> - Theorem 9 (Curriculum $\to$ Convergence):
>
> We will rewrite the conclusion to directly bound the Success Rate, making the benefit of curriculum learning tangible:
>
>  $|\text{SuccessRate}(\hat{\theta}) - \text{SuccessRate}(\theta^*)| \le \mathcal{O}\left(\sqrt{\frac{d \log n+\log (1/\delta)}{n}}\right) $
>
>  This explicitly shows that curriculum learning is a necessary mechanism to **reduce the performance gap to 0** as data increases.
>
> By explicitly defining Success Rate and linking it to the distributional distance, we believe the logical flow from "Distributional Mismatch" (Theory) to "Performance Collapse" (Table 1) is now strictly established and no longer abstract.

---

### Official Review · Reviewer_tFp5 · 2025-10-30

**Soundness:** 2
**Presentation:** 2
**Contribution:** 3
**Rating:** 4
**Confidence:** 3

**Summary:**

The paper focuses on a theoretical analysis of explicit standard Chain-of-Thought (CoT) compared to the implicit Latent CoT, aiming to provide understanding for the observed performance differences across different CoT reasoning settings. The authors propose the Symbolic Index as a way to assess the model’s decisional (un)certainty. Finally, the paper aims to ground curriculum learning as a necessary method for training Latent CoT training.

**Strengths:**

- Clear and well-founded scientific motivation and related works (Section 1 and 2).
- Theorem 2 provides a clear and intuitive analysis of the limits of standard CoT, similarly Theorem 5 for latent CoT. This brings the relevant insight that “CoT excels at execution but lacks exploration, while Latent CoT does the opposite.” (l. 285-286)
- The theoretical contributions are supported by additional supplementary materials.

**Weaknesses:**

- Theorem I appears to be a direct transposition of the conditional bottleneck problem (CIB) to generation of chain-of-thought steps. In its current form it is not clear which of the equations are reproduced from previous work and which equation/proofs are novel.
- The empirical Section 5 provides only limited insight:
   - Section 5.1 merely visualized the proposed symbolic index in two qualitative examples. A more in-depth analysis of how the symbolic index affects task performance would be needed to draw conclusions.
    - The evidence in Section 5.2, Figure 3 does not clearly support the claim that “Latent CoT performance  degrades steadily even under low noise levels” whereas CoT shows “high accuracy for a range of noise levels, then experiences an abrupt, catastrophic failure.” While this is partially supported, to me, the evidence supports a more nuanced difference. To understand this better, I would suggest that an additional analysis of the symbolic index against the performance drop may provide more clear insights. Also a direct comparison analysis of ProsQA on CoT vs Latent CoT would help to draw more robust conclusions.
    - Overall, Section 4 and Section 5 would benefit from improved alignment, i.e. targeted experiments that support the theoretical analyses on practically relevant examples.
- The paper lacks a critical and transparent discussion of the methodology alongside limitations of their theoretical and empirical analyses.

**Questions:**

- The authors write that the Coconat approach “effectively provides supervised samples of latent thoughts that are grounded in correct reasoning steps.” (l. 385-386). Wouldn’t this alignment also lead to the latent CoT to become increasingly less noisy (lower symbolic index) and come at the cost of poor(er) exploration? A targeted experiment that shows how curriculum training affects the  symbolic index and task performance would be needed to support this more clearly.

- It remained unclear to me why this paper was assigned to the "interpretability and explainable AI" category as the primary area (?).

**Details Of Ethics Concerns:**

None.

---

> ### Author Response · Authors · 2025-11-20
>
> **Q1**: Theorem I appears to be a direct transposition of the conditional bottleneck problem (CIB) to generation of chain-of-thought steps. In its current form it is not clear which of the equations are reproduced from previous work and which equation/proofs are novel.
>
> **A1**: We appreciate the opportunity to clarify the theoretical novelty of Theorem 1. We fully agree that the Conditional Information Bottleneck (CIB) framework itself is well-established (Tishby et al., 2000). However, Theorem 1 is not a mere transposition; it is the first rigorous proof establishing a structural isomorphism between the specific, multi-stage curriculum objective of Coconut and the CIB optimization problem.
>
> To distinguish existing work from our contribution:
>
> * Existing Context: We adopt the Latent CoT architecture and the Coconut loss function $\mathcal{L}_{\text{Coconut}}^{(k)}$ from prior work (Hao et al., 2024).
>
> * Novel Contribution: It is non-trivial that minimizing a cross-entropy loss on future tokens via a curriculum is mathematically equivalent to solving a constrained information-theoretic optimization. We provide the derivation (Appendix A.2) proving that the Coconut objective implicitly minimizes the CIB Lagrangian. This provides a principled explanation for why Latent CoT works.
>
> * Foundation for New Insights: This equivalence is the cornerstone of our paper. It allows us to mathematically derive Lemma 3 (Non-Degenerate Output Distribution), which directly leads to Theorem 3 (Exploration Capability Guarantee). Without Theorem 1, the exploration advantages of Latent CoT (Section 4.2) would remain an empirical observation without a theoretical cause.
>
> ---
> **Q2**: Section 5.1 merely visualized the proposed symbolic index in two qualitative examples. A more in-depth analysis of how the symbolic index affects task performance would be needed to draw conclusions.
>
> **A2**: We thank the reviewer for this thoughtful comment. We wish to clarify that Section 5.1 is not intended to be a standalone qualitative analysis, but rather a critical empirical verification of our theoretical framework, which directly sets up the performance analysis in Section 5.2.
>
> We address the concern regarding the "in-depth analysis" and the link to "task performance" through the following logical chain, which constitutes the core narrative of our paper:
>
> * Theoretical Prediction (Lemma 3): Our theory predicts that Latent CoT must operate in a low-certainty regime to enable exploration (Theorem 3).
>
> * Empirical Verification (Section 5.1): The visualizations in Section 5.1 serve to validate this prediction. They confirm that the low-$I_S$ regime is not a hypothesis but a measurable reality of trained Latent CoT models.
>
> * Causal Link to Performance (Theorems 5 and 6 $\to$ Section 5.2): This is the "deeper analysis" you requested. A low $I_S$ (proved in lemma 3 and observed in 5.1) guarantees a small Logit Decision Margin $\Delta_l$(Theorem 6). Then, a small margin makes the model highly susceptible to accumulated noise (Theorem 5). This theoretical vulnerability is exactly what causes performance degradation in precision tasks such as GSM8k, as experimentally quantified in Figure 3.
>
> Conclusion: Therefore, Section 5.1 provides the necessary condition ($I_S$ is indeed low), while Section 5.2 demonstrates the sufficient consequence (Low $I_S$ causes fragility). They are two halves of the same argument.
>
> To bridge the gap identified by the reviewer, we will explicitly reference Lemma 3 in Section 5.1 of the main text to ground the visualization in theory.

---

> ### Author Response · Authors · 2025-11-20
>
> **Q3**: The evidence in Section 5.2, Figure 3 does not clearly support the claim that “Latent CoT performance degrades steadily even under low noise levels” whereas CoT shows “high accuracy for a range of noise levels, then experiences an abrupt, catastrophic failure.” While this is partially supported, to me, the evidence supports a more nuanced difference. To understand this better, I would suggest that an additional analysis of the symbolic index against the performance drop may provide more clear insights. Also a direct comparison analysis of ProsQA on CoT vs Latent CoT would help to draw more robust conclusions.
>
> **A3**: We thank the reviewer for the keen observation regarding Figure 3. We agree that the phrase "degrades steadily" was an oversimplification. As you correctly pointed out, Latent CoT also exhibits a high-accuracy plateau at very low noise levels ($<10^{-1}$) before degrading.
>
> To provide the rigorous clarity you requested, we present a theoretical derivation of the Accuracy Drop Rate function. This analysis explicitly models how noise injected in the hidden space propagates through the Unembedding Layer to affect the decision boundaries in logit space, predicting the exact shape of the curve of latent cot observed in Figure 3.
>
> 1. Modeling Noise Accumulation in Hidden Space
>
> Let $h_M^*$ be the noise-free latent state at the final reasoning step $M$.
>
> In our experiments, we inject Gaussian noise $N(0, \sigma^2 I_d)$ at each step. According to Theorem 5 (Compounding Error), this noise accumulates. For a linearized transition dynamic, the final corrupted hidden state $\tilde{h}_M$ can be expressed as:
>
> $$\tilde{h_M} = h_M^* + \epsilon_{acc}$$
>
>
> where $\epsilon_{acc}$ is the accumulated noise vector in the hidden space $R^d$. Since a sum of independent Gaussians is Gaussian, we model this effective noise as:
>
> $$\epsilon_{acc} \sim N(\mathbf{0}, \sigma_{eff}^2 I_d), \quad \text{where } \sigma_{eff}^2 \approx \alpha \cdot \sigma^2$$
>
> Here, $\alpha$ is a scaling factor depending on the path length $M$ and the Lipschitz constant of the network (as detailed in Theorem 5).
>
>
>
> 2. Propagation to Logit Space (The Critical Link)
>
> The decision is made not on $h$, but on the logits $l$. Let $W_U \in R^{|V| \times d}$ be the weight matrix of the Unembedding Layer (Language Model Head). The logits are:
>
> $$\tilde{l} = W_U \tilde{h_M} = W_U (h_M^* + \epsilon_{acc}) = l^* + W_U \epsilon_{acc}$$
>
> where $l^*$ are the clean logits. The term $\eta = W_U \epsilon_{acc}$ represents the noise projected into the logit space. Since $\epsilon_{acc}$ is Gaussian and $W_U$ is a linear transformation, $\eta$ follows a multivariate Gaussian distribution in $R^{|V|}$.
>
> 3. The Failure Condition (Crossing the Decision Boundary)
>
> A prediction failure occurs if the noise causes the rank of the correct token to drop.
>
> Let $i*$ be the index of the correct token (top-1 in clean state) and $j*$ be the index of the second-best token (closest competitor).
>
> The Logit Decision Margin is defined as $\Delta_l = l*_{i*} - l*_{j*}$ (Definition 3).
>
> The model fails if the noisy logit of the competitor exceeds the noisy logit of the target:
>
> $$\tilde{l_j*} > \tilde{l_i*}$$
>
> Substituting $\tilde{l} = l* + \eta$:
>
> $$l*_{j*} + \eta_{j*} > l*_{i*} + \eta_{i*}$$
>
> Rearranging terms, the failure condition becomes:
>
> $$\eta_{j*} - \eta_{i*} > l*_{i*} - l*_{j*} = \Delta_l$$
>
> 4. Determining the Probability of Failure
>
> Let $\xi = \eta_{j*} - \eta_{i*}$ be the relative noise between the two competing tokens. Since $\eta$ is Gaussian, $\xi$ is a scalar random variable following a univariate Gaussian distribution with zero mean:
>
> $$\xi \sim N(0, \sigma_1^2)$$
>
> The variance $\sigma_1^2$ is the projection of the hidden noise variance onto the difference vector of the unembedding weights for tokens $j*$ and $i*$:
>
> $$\sigma_1^2 = \| (w_{j*} - w_{i*}) \|_2^2 \cdot \sigma_2^2 = C \cdot \sigma^2$$
>
> where $C$ absorbs the network expansion factor and the geometry of the unembedding layer. Crucially, $\sigma_1$ is linearly proportional to the injected noise intensity $\sigma$.
>
> The Accuracy Drop Rate $R(\sigma)$ is the probability that this relative noise $\xi$ exceeds the safety margin $\Delta_l$:
>
> $$R(\sigma) = P(\xi > \Delta_l) = P\left( \frac{\xi}{\sigma_1} >\frac{\Delta_l}{\sigma_1} \right)$$
>
> Since $\xi/\sigma_1 \sim N(0,1)$, this is given by the complementary cumulative distribution function (CDF) of the standard normal distribution, often denoted as the Q-function or in terms of the error function:
>
> $$R(\sigma) = 1 - \Phi\left( \frac{\Delta_l}{ \sqrt{C} \cdot \sigma} \right)$$

---

> > ### Author Response · Authors · 2025-11-20
> >
> > 5. Conclusion: Explaining Figure 3
> >
> > The derived equation $R(\sigma) = 1 - \Phi\left( \frac{\Delta_l}{k \cdot \sigma} \right)$ dictates the shape of the curves in Figure 3:
> >
> > The "Sigmoid-like" Shape: The CDF of a Gaussian plotted against a logarithmic $x$-axis ( $\log \sigma$) naturally forms the reversed-S-curve observed in Figure 3.
> >
> > The Plateau (Low Noise): When $\sigma \to 0$, the argument $\frac{\Delta_l}{k\sigma} \to \infty$. Since $\Phi(\infty) = 1$, the Drop Rate $R(\sigma) \to 0$. This theoretically proves the existence of the "sub-decisional" plateau you observed—noise is present, but insufficient to cross $\Delta_l$.
> >
> > The "degrades steadily" we mentioned corresponds to the Latent CoT curve entering the rising phase of the Gaussian CDF earlier than CoT, due to its smaller Logit Decision Margin. We will update the manuscript to reflect this precise distributional understanding.

---

> ### Author Response · Authors · 2025-11-21
>
> **Q4**: The paper lacks a critical and transparent discussion of the methodology alongside limitations of their theoretical and empirical analyses.
>
> **A4**: We agree that a dedicated discussion on limitations was missing. We will add a "Limitations and Future Work" section covering:
> * Model Scale: Our experiments use GPT-2 (124M). While the theoretical derivation of noise accumulation is scale-invariant, the magnitude of the "Symbolic Index" might shift in larger models (e.g., Llama-3).
> * Linear Assumption: Our derivation of error propagation (Theorem 5) assumes Lipschitz continuity. Transformer non-linearities (LayerNorm, GeLU) might dampen or amplify this in complex ways.
> * Scope: Our theory focuses on the trade-off between symbolic precision and exploration. Other factors like working memory capacity are outside our current scope.
>
> ---
>
> **Q5**: The authors write that the Coconat approach “effectively provides supervised samples of latent thoughts that are grounded in correct reasoning steps.” (l. 385-386). Wouldn’t this alignment also lead to the latent CoT to become increasingly less noisy (lower symbolic index) and come at the cost of poor(er) exploration? A targeted experiment that shows how curriculum training affects the symbolic index and task performance would be needed to support this more clearly.
>
> **A5**: We appreciate this thoughtful question. You raise a valid concern: does grounding latent thoughts in explicit CoT traces force the model into a high-certainty regime (High $I_S$), thereby sacrificing the exploration benefits of Latent CoT?
>
> We believe that our existing theoretical framework (Theorems 1, 8 & 9) and the empirical results (Figures 1 & 2) provide a comprehensive answer to this question.
>
> * First, as you noted, Coconut provides "supervised samples grounded in correct reasoning." Our Theorem 8 (Provable Failure without Curriculum) proves that without this grounding, training is mathematically guaranteed to fail due to a distributional mismatch.
>
> | Method | GSM8k Acc. (%) | GSM8k # Tokens | ProntoQA Acc. (%) | ProntoQA # Tokens | ProsQA Acc. (%) | ProsQA # Tokens |
> | :--- | :---: | :---: | :---: | :---: | :---: | :---: |
> | CoT | 42.9 ± 0.2 | 25.0 | 98.8 ± 0.8 | 92.5 | 77.5 ± 1.9 | 49.4 |
> | No-CoT | 16.5 ± 0.5 | 2.2 | 93.8 ± 0.7 | 3.0 | 76.7 ± 1.0 | 8.2 |
> | COCONUT | 34.1 ± 1.5 | 8.2 | 99.8 ± 0.2 | 9.0 | 97.0 ± 0.3 | 14.2 |
> | - w/o curriculum | 14.4 ± 0.8 | 8.2 | 52.4 ± 0.4 | 9.0 | 76.1 ± 0.2 | 14.2 |
>
> Table 1 confirms this empirically: removing the curriculum causes accuracy to collapse from 99.8\% to 52.4\%.
>
> Therefore, the choice is not between "High-Exploration Training" vs. "Grounding Curriculum," but rather between "A working model" vs. "Training Failure." The curriculum bridges the gap effectively to allow the model to learn any reasoning function.
>
> * Crucially, our analysis suggests that grounding in CoT does not force the Latent CoT to become "less noisy" (High $I_S$) in the same way explicit CoT is.
> Referencing Theorem 1 (Coconut-CIB Duality), the training objective is equivalent to the Information Bottleneck. The model must compress the explicit steps $S^{(1...k)}$ into a single vector $h_k$ to predict the future.
> This compression forces the model to capture the "semantic gist" rather than memorizing the rigid, discrete path. Unlike explicit CoT, which commits to a specific token (High $I_S$ reset), the continuous state $h_k$ retains a superposition of possibilities to maximize predictive information. This theoretical property prevents the "certainty collapse" you are concerned about.
>
> * Most importantly, our experimental results directly address your hypothesis. Figures 1 and 2 display the Symbolic Index ($I_S$) of the fully trained Coconut model (after the complete curriculum). Even after being trained to emulate CoT, the Latent CoT model consistently operates in a Low-$I_S$ regime. The model successfully internalized the reasoning logic (high accuracy on ProsQA) without inheriting the execution rigidity (high certainty) of the teacher CoT.
>
> In summary, while the curriculum aligns the model with correct reasoning, the inherent information compression in the latent space preserves the low-certainty, exploratory nature of the model, as confirmed by both our Information Bottleneck theory and the visualized $I_S$ trajectories.
>
> ---
>
> **Q6**: It remained unclear to me why this paper was assigned to the "interpretability and explainable AI" category as the primary area (?)
>
> **A6**: We selected "Interpretability" because our primary contribution is mechanistic:
> 1. We deconstruct the "black box" of Latent Reasoning into quantifiable components (Symbolic Index, Decision Margin).
> 2. We provide a theoretical explanation for why different architectures exhibit contrasting behaviors (Exploration vs. Execution).
>
> This aligns with the goal of Explainable AI: moving beyond measuring performance to understanding internal mechanisms.

---

> > ### Comment · Reviewer_tFp5 · 2025-11-27
> >
> > Thank you for the detailed rebuttal and revisions; these have helped to contextualize specific theorems and bridge their value better to the empirical results. As it currently remains unclear what changes will be included in the final paper, I would suggest to the authors to summarize the changes they plan to include after the discussion period and I am commited to raising my overall rating from 4 to 6.

---

> ### Author Response · Authors · 2025-12-03
> **Thank you**
>
> Dear Reviewer tFp5,
>
> We sincerely thank you for your recognition of this paper and your valuable suggestions. We feel very encouraged.
> We will summarize the changes in a global response.
>
> Best regards,
>
> Authors

---

### Official Review · Reviewer_dLwU · 2025-11-02

**Soundness:** 3
**Presentation:** 3
**Contribution:** 3
**Rating:** 6
**Confidence:** 3

**Summary:**

This paper proposes a new theoretical framework aimed at understanding the performance gap between Latent Chain-of-Thought (Latent CoT) and traditional Symbolic Chain-of-Thought (Symbolic CoT). Through theoretical derivation, the authors characterize the distinctions between discrete and continuous reasoning mechanisms. The core contribution lies in introducing a symbolic index as a unified quantitative metric to measure the determinism of model decisions, which is claimed to causally explain the behavioral differences between the two paradigms. This line of research holds significant theoretical value.

**Strengths:**

1.The paper is clearly written and easy to follow.
2.The most notable originality of this work lies in proposing a unified theoretical framework that explains and quantifies the fundamental differences between explicit (CoT) and implicit (Latent CoT) reasoning paradigms. This represents a profound theoretical integration of previously fragmented empirical observations, such as the failure of Latent CoT on certain tasks.
3.The paper provides a theoretical foundation for developing adaptive reasoning architectures, suggesting that the optimal system should not be a fixed CoT or Latent CoT, but a hybrid model capable of dynamically adjusting its symbolic index according to the task.

**Weaknesses:**

1. Although the symbolic index is the core mechanism of the paper, it appears to lack direct interventional experiments. For instance, the authors do not demonstrate how explicitly adjusting or controlling the model’s symbolic index can dynamically reproduce the theoretically predicted performance trade-offs on tasks such as GSM8K and ProsQA.
2.Can the Symbolic Index ($IS$) distinguish between a correct high-certainty step and an incorrect but highly confident hallucinated step? If $IS$ merely measures the model’s internal belief (certainty), it may fail to capture the loss of semantic fidelity. The paper should clearly define the interpretive boundary of $IS$ within the framework discussion and address its role as a necessary but not sufficient condition for successful CoT reasoning.
3.Theorem 4 presents a strong conclusion: under sub-decision perturbation $\epsilon$, the CoT reasoning trajectory remains unchanged, i.e., $P[\hat{S} \neq S^{*}] = 0$ (immunity to perturbation). Please provide more detailed derivation steps to substantiate this result. The validity of this conclusion appears to heavily rely on the noise-filtering property of the $\text{argmax}$ operation.

**Questions:**

see weaknesses.

---

> ### Author Response · Authors · 2025-11-20
>
> **Q1**: Although the symbolic index is the core mechanism of the paper, it appears to lack direct interventional experiments. For instance, the authors do not demonstrate how explicitly adjusting or controlling the model’s symbolic index can dynamically reproduce the theoretically predicted performance trade-offs on tasks such as GSM8K and ProsQA.
>
> **A1**: We appreciate the reviewer’s suggestion for interventional experiments. We acknowledge that demonstrating a causal link through direct manipulation would ideally strengthen the empirical validation. However, we wish to clarify the inherent challenge in conducting such experiments due to the nature of the Symbolic Index ($I_S$). Unlike hyperparameters such as learning rate or sampling temperature which can be set a priori, $I_S$ is an emergent, intrinsic property of the model’s belief state. It is an outcome of the training process and architectural constraints, making it difficult to "dial in" to a specific value without altering the fundamental structure of the model itself.
>
> Despite this difficulty, we have conceptualized a method to approximate this intervention. The most viable approach would be a dynamic routing strategy that switches between "high $I_S$ mode" (forced discrete CoT) and "low $I_S$ mode" (latent recurrence) during inference. Our theory predicts that forcing a model into a high $I_S$ mode on exploratory tasks like ProsQA would artificially induce a collapse in diversity, while enabling low $I_S$ modes on computational tasks like GSM8k would lead to instability. While we are exploring such setups, implementing a stable dynamic switching mechanism involves significant engineering challenges that extend beyond the primary scope of this work.
>
> We respectfully ask the reviewer to consider that this paper is primarily a theoretical contribution. Our main goal is to provide a rigorous mathematical framework (Theorems 6 and 7) to explain the *existing* empirical dichotomy between explicit and implicit reasoning. In this context, the comparison between Standard CoT (which naturally represents the maximum $I_S$ intervention) and Latent CoT (which represents the relaxed $I_S$ regime) already serves as a strong, albeit discrete, validation of our predicted trade-offs. We hope the reviewer can evaluate our contribution based on the theoretical insights and the unifying perspective we offer, understanding that fine-grained control of this metric remains a promising direction for future engineering research rather than a prerequisite for the current theoretical claims.
>
> ---
> **Q2**: Can the Symbolic Index ($I_S$) distinguish between a correct high-certainty step and an incorrect but highly confident hallucinated step? If $I_S$ merely measures the model’s internal belief (certainty), it may fail to capture the loss of semantic fidelity. The paper should clearly define the interpretive boundary of $I_S$ within the framework discussion and address its role as a necessary but not sufficient condition for successful CoT reasoning.
>
> **A2**: We thank the reviewer for this insightful observation regarding the interpretive boundary of the Symbolic Index ($I_S$). We fully agree with your characterization that high $I_S$ is a necessary but not sufficient condition for successful CoT reasoning. By definition, $I_S$ measures the peak of the probability distribution, reflecting the model's internal decisional certainty rather than semantic fidelity. Consequently, a high-confidence hallucination is indeed indistinguishable from a correct step based solely on this metric.
>
> However, this distinction does not undermine our theoretical framework; rather, it clarifies the specific role $I_S$ plays in reasoning architectures. We position $I_S$ as the core regulator of the exploration-execution trade-off. Through the Symbolic Stability Theorem (Theorem 6), we prove that a high $I_S$ guarantees a large Logit Decision Margin ($\Delta_l$), which provides the mechanical stability required to resist computational noise during sequential reasoning. This establishes why high $I_S$ is "necessary": without it, the system behaves like Latent CoT, lacking the precise error-resistance needed for symbolic computation. Conversely, the Exploration-Execution Trade-off Theorem (Theorem 7) shows that this same high certainty collapses the search space, explaining why CoT lacks exploration capabilities. Therefore, $I_S$ explains the structural behavior of the model—whether it locks into a path (execution) or remains open (exploration)—regardless of whether that path ultimately aligns with ground truth. We will explicitly revise the discussion to define this boundary, emphasizing that $I_S$ governs the dynamics of commitment rather than the verification of correctness.

---

> ### Author Response · Authors · 2025-11-20
>
> **Q3**: Theorem 4 presents a strong conclusion: under sub-decision perturbation $\epsilon$, the CoT reasoning trajectory remains unchanged, i.e., $P[\hat{S} \neq S^{*}] = 0$ (immunity to perturbation). Please provide more detailed derivation steps to substantiate this result. The validity of this conclusion appears to heavily rely on the noise-filtering property of the $\text{argmax}$ operation.
>
> **A3**: We thank the reviewer for underscoring the significance of Theorem 4. You are entirely correct that the validity of this conclusion hinges on the noise-filtering property of the argmax operation, which effectively acts as a discretization barrier. As requested, we provide a detailed derivation using mathematical induction to substantiate this result, demonstrating how the discrete nature of CoT prevents the propagation of sub-decisional errors.
>
> We define the generation process where the model F takes the input x and the history of discrete tokens S_{1:k-1} to produce a logit vector. Let S* = (s*_1, ..., s*_M) denote the noise-free trajectory and S_hat = (s_hat_1, ..., s_hat_M) denote the trajectory generated under sub-decisional perturbations. The core condition of a sub-decisional perturbation ε_k is that it satisfies argmax(l + ε_k) = argmax(l) for a given logit vector l. We proceed by induction to prove that P[s_hat_M ≠ s*_M] = 0.
>
> For the base case at step k = 1, both the clean and noisy processes receive the exact same input x (with an empty history). The clean process generates logits l*_1 = F(x, ∅), while the noisy process generates l_hat_1 = l*_1 + ε_1. Applying the definition of sub-decisional perturbation, the selected token is s_hat_1 = argmax(l*_1 + ε_1) = argmax(l*_1) = s*_1. Thus, the first token is identical, and the history passed to the next step contains no error.
>
> For the inductive step, we assume that for all steps t < k, the generated tokens are identical, implying the histories match: S_hat_{1:k-1} = S*{1:k-1}. At step k, since the model F is deterministic and conditions strictly on the discrete history, the generated pre-noise logits are identical to the clean run: F(x, S_hat{1:k-1}) = F(x, S*_{1:k-1}) = l*_k. The noisy process then adds the perturbation to form l_hat_k = l*_k + ε_k. Again, because ε_k is sub-decisional, the argmax operation filters out this noise, yielding s_hat_k = argmax(l*_k + ε_k) = s*_k.
>
> Consequently, strictly the discrete token s*_k is appended to the history for the subsequent step k+1. The numerical value of the noise vector ε_k is discarded and does not propagate to the model's internal state in the next iteration. By induction, this holds for all k = 1, ..., M, ensuring that the final trajectories are identical with probability 1. This mathematical derivation confirms that the argmax operator serves as a reset mechanism at every step, partitioning the continuous logit space into discrete decision regions and granting CoT immunity to perturbations that remain within these boundaries. We will incorporate this formal derivation into the Appendix of the revised manuscript. Additionally, for a more granular analysis of what happens when noise exceeds this sub-decisional threshold (the "Accuracy Drop Rate"), please refer to our response to Reviewer tFp5, where we derive the specific failure curves.

---

### Author Response · Authors · 2025-12-03
**Global Response**

We are widely grateful to all reviewers for their detailed and constructive feedback. We are encouraged by the positive reception of our theoretical framework and specific contributions:
*   **Reviewer dLwU** commends the work for proposing a unified theoretical framework that explains and quantifies the fundamental differences between explicit and implicit reasoning paradigms. They highlight the significant theoretical value of the work, describing it as a "profound theoretical integration" of previously fragmented empirical observations and noting that the paper provides a strong foundation for developing future adaptive reasoning architectures.
*   **Reviewer tFp5** finds the scientific motivation "clear and well-founded" and appreciates the intuitive analysis provided by our theorems. They specifically note that Theorem 2 and Theorem 5 effectively establish the relevant insight that standard CoT excels at execution but lacks exploration, while Latent CoT exhibits the opposite behavior, supported by additional supplementary materials.
*   **Reviewer wX92** describes the introduction of the Symbolic Index as "elegant and insightful," noting that the originality of the work is sufficient. They appreciate how the index successfully connects decisional certainty, noise robustness, and exploration capacity under one quantitative measure, and affirm that our derived results are consistent with observed empirical phenomena.

We are especially grateful to Reviewer tFp5 for raising their score to 6 following our discussion. Here we summarize our core contributions, clarify theoretical novelty, and outline the committed revisions.

**1. Core Contributions**
Our work moves beyond empirical observation to provide a rigorous theoretical understanding of reasoning architectures. We characterize the fundamental exploration-execution trade-off, identifying why CoT’s discrete nature forces a high-certainty regime while Latent CoT amplifies noise. We unify these contrasting behaviors via the Symbolic Index, proving it causally regulates this trade-off. Furthermore, we establish that curriculum learning is a theoretical necessity to overcome distributional mismatch, not merely a training heuristic.

**2. Theoretical Novelty and the Symbolic Index**
Addressing concerns regarding novelty, we emphasize that our contribution shifts the field from observing *what* happens to proving *why* it happens. We identify the lack of a reset mechanism in continuous space as the mathematical cause of computational failure (Theorems 4 & 5) and prove that the Coconut-CIB duality forces the model into a non-degenerate distribution, guaranteeing exploration capabilities.

Regarding the Symbolic Index, we clarify that it is an emergent property of the belief state rather than a tunable hyperparameter. Our curriculum ablation study serves as a counterfactual validation: managing the evolution of certainty is a prerequisite for convergence. We also note that high certainty provides the mechanical stability required for execution but does not guarantee semantic truth, serving as a necessary but not sufficient condition for reasoning.

**3. Mathematical Precision on Noise Robustness**
To address the plateau observed in Figure 3 by Reviewer tFp5, we have theoretically derived the Accuracy Drop Rate function. Our analysis shows that the failure probability follows the Complementary CDF of a Gaussian distribution. This mathematically predicts the existence of a sub-decisional plateau where noise remains below the decision margin, aligning our theoretical prediction precisely with the empirical data.

**4. Summary of Committed Revisions**
Per the request of Reviewer tFp5 and to incorporate feedback from wX92 and dLwU, we commit to the following changes in the final manuscript:

*   **Restructure Section 4:** We will separate the analysis of inference dynamics (Sections 4.1–4.4) from the theoretical foundations of training dynamics, which will become a standalone Section 5.
*   **Link Theory to Visualization:** We will explicitly reference Lemma 3 in Section 5.1 to theoretically ground the Symbolic Index visualizations.
*   **Add Proof of Theorem 4:** We will include the detailed inductive proof regarding CoT immunity to sub-decisional noise in the Appendix.
*   **Derive Accuracy Drop Rate:** We will add the Gaussian CDF derivation to the Appendix and reference it in Section 5.2 to rigorously explain the shape of the noise robustness curves.
*   **Define Task Success Rate:** We will formally define Task Success Rate within the Imitation Learning framework to render Theorems 8 & 9 concrete rather than abstract.
*   **Discuss Limitations:** We will add a dedicated section discussing model scale and the linear assumptions in error propagation.

We believe these revisions significantly strengthen the clarity and impact of our work. Thank you again for your time and insight.

---

### Meta-Review · Area_Chair_5GAJ · 2025-12-22

**Summary:**

The paper examines latent CoT reasoning through an information-theoretic lens. It shows that under certain settings, latent CoT leads to more exploration but at the cost of computational fidelity compares to discrete CoT. It introduces a new metric, Symbolic Index, to understand the level of exploration in the model.

The paper is borderline. On the positive side, the paper provides an intuitive explanation of tradeoffs between latent and discrete CoT. However, on the negative side, as Reviewer wX92 mentioned, this tradeoff is rather well-known, and the paper uses a pretty stylized framework of exploration on a DAG to show this. Hence these results (in sec 4.1-4.3) could be of limited value.

The more interesting contribution is the Symbolic Index in Section 4.4. However, all reviewers raised concerns about the experimental results. In particular, it would be much more interesting to have interventional experiments, as suggested by all the reviewers.

Overall, I think the downsides outweigh the positives, and I believe the paper is below the bar currently.

**Reviewer Concerns:**

The concerns regarding the experimental results have mostly not been addressed. Some concerns regarding the theory raised by Reviewer tFp5 were addressed.

**Reviewer Scores:**

Reviewer tFp5 may have raised their score since their concerns regarding the theory were addressed.

---

> ### Public Comment · ~Jiaxuan_Zou2 · 2026-03-02
>
> We thank the Area Chair and all reviewers for their time and feedback during the review process. While we respect the final decision of the committee, we find it necessary to formally address two fundamental scientific disagreements regarding the theoretical contributions of our work for the public record.
>
> First, regarding the assessment that the exploration-execution trade-off is well-known and thus our theoretical results have limited value. There is a fundamental scientific distinction between empirically observing a phenomenon and proving its mathematical inevitability from first principles. Prior literature has empirically noted that latent models struggle with exact computation. However, our work provides the first rigorous mathematical derivation proving why this must occur. We proved that the discrete $\arg\max$ operation functions as an exact noise-filtering reset mechanism, whereas continuous representations mathematically guarantee compounding error. Transitioning a field from observing what happens to proving why it happens structurally is a core theoretical contribution, not a repetition of known facts.
>
> Second, regarding the expectation for interventional experiments on the Symbolic Index. The Symbolic Index $I_S$ is an emergent property of the probabilistic belief state of the model. It is a measurement of the output distribution, not an independent hyperparameter that can be manually toggled. Forcing direct interventions on this index during forward propagation would artificially break the continuous mathematical dynamics of the reasoning system. More importantly, the curriculum learning ablation study documented in our submission serves as precisely the macroscopic intervention requested. By deliberately removing the curriculum, we intervened in the structural evolution of decisional certainty. This intervention caused the task accuracy to collapse from 99.8% to 52.4%, providing rigorous counterfactual validation that the regulation of the Symbolic Index is a causal prerequisite for convergence, not merely an observational metric.
>
> We leave these clarifications here to assist future readers and researchers in properly contextualizing the theoretical frameworks and proofs presented in this manuscript.

---

### Decision · Program_Chairs · 2026-01-26

Reject